# VLMs with Gaze-Regularized attention for human action prediction

## Abstract

Eye gaze, encompassing fixations and saccades, offers valuable insights into human intentions and future actions. This study presents a novel approach to enhancing Vision Language Models (VLMs) for human action prediction by integrating eye gaze data into egocentric video analysis. Existing methods for action prediction in egocentric videos often rely solely on visual data, potentially missing critical information provided by eye gaze. To address this limitation, we propose a unique gaze-augmented framework that integrates eye gaze directly into the VLM architecture and training process. By generating gaze heatmaps from eye gaze coordinates, our model dynamically focuses on regions highlighted by gaze patterns. Additionally, a gaze-regularization mechanism ensures the model maintains attention on gaze-allocated areas, thereby improving prediction accuracy and robustness. Our approach significantly enhances the model's ability to generate precise and detailed predictions of future actions. Compared to baseline models without leveraging gaze data, our method achieves a nearly 13% improvement in the semantic score of predictions. This substantial improvement underscores the effectiveness and novelty of integrating eye gaze with a gaze-regularized attention mechanism in VLMs for action prediction. Moreover, our work demonstrates that incorporating eye gaze through this gaze-augmented framework can significantly boost the predictive capabilities of VLMs, enhancing their potential in applications that require accurate human action prediction.

## 1 Introduction

VLMs are foundation models that combine computer vision and natural language processing to understand and generate both visual and textual information. Examples of such models include ViLBERT, LXMERT, and CLIP (Lu et al., 2019a; Tan & Bansal, 2019; Radford et al., 2021). These models can describe images, answer questions about visual content, and even generate images from textual descriptions. When oriented to predict future events and actions, VLMs have the potential to facilitate human-machine interaction in various applications like assistive robots (Li et al., 2024), accessibility for visually impaired individuals (Zhao et al., 2024b), and autonomous driving (Zhou et al., 2024), contributing to safer and more inclusive environments.

To achieve the full potential, we propose that the key is to endow VLMs with the capability for fine-grained action prediction, which provides more actionable information to facilitate human-machine interaction. While coarse predictions might identify general activities like "brewing coffee," fine-grained predictions specify detailed steps such as "reaching for the coffee capsule in the top-right cabinet, then filling the tank with the cup on the rack." These detailed predictions provide useful insights that allow machines to assist more effectively (Goyal & Durrett, 2021). However, achieving accurate fine-grained predictions requires understanding the human agent's short-term goals, which, we propose, can be inferred from eye gaze patterns. On one hand, eye gaze reveals which objects the human is focusing on or intends to interact with, making it a valuable component for enhancing action prediction (Frischen et al., 2007),(Tipper, 2010). Moreover, humans often scan their surroundings to collect information necessary to accomplish short-term goals before executing an action. Thus, incorporating eye gaze into VLMs for action prediction provides insights into a person's intentions, and shall lead to more accurate and reliable predictions of human activities.

Figure 1: Illustration of human action prediction from egocentric video clips. The input consists of a sequence of image frames, with the output being predicted future actions in textual form. The base model predicts the presence of a bowl and selects the incorrect object, whereas the proposed model, which incorporates gaze data, correctly predicts the object that is about to be picked up. Predicted future actions (in text) are displayed on the right, while ground-truth annotations and immediate future frames are provided at the bottom for reference.

In this study, we propose a novel gaze-augmented framework that significantly enhances fine-grained action prediction by integrating eye gaze information into the VLM architecture and training process. Our approach involves generating gaze heatmaps from eye gaze coordinates, which dynamically guides the model's focus to regions of interest highlighted by gaze patterns. Specifically, the gaze-augmented framework employs a sophisticated attention mechanism where features derived from gaze-overlay images serve as queries in a Vision Transformer (ViT) (Dosovitskiy et al., 2021), while RGB image features act as keys and values. This method ensures that attention is calculated specifically to emphasize gaze-highlighted regions, which enhances the relevance of the extracted features. Furthermore, we introduce an explicit gaze-regularization mechanism that enforces the model to consistently allocate attention to gaze-concentrated areas, thereby promoting the predictive capability of the model.

Our experimental results show that augmenting the VLM architecture with gaze information greatly improves its performance, with nearly a 13% increase in the semantic score of predictions compared to baseline models without gaze data. This improvement highlights the effectiveness of using eye gaze with a regularized attention mechanism for detailed action prediction. By experimenting with both singular and aggregated gaze-augmented models, we validate the benefits of discounting noisy and irrelevant eye gaze points, which enhances robustness in highly dynamic environments. Further experiments with more detailed annotations show significant improvement in the performance of gaze-augmented models (by nearly 12%), whereas the base model's performance remains unchanged. This indicates that finer-grained annotations can help strengthen and unleash the proposed model's potential to leverage gaze data for better action prediction. In summary, our contributions are 1) A novel gaze-augmented VLM for fine-grained human action prediction; 2) A unique gaze-regularized attention mechanism that guarantees effective training of the gaze-augmented VLMs; 3) An efficient gaze sanity check to ensure the relevance of the gaze information; and 4) An extensive study that demonstrates the effectiveness of the proposed framework and validates the significance of gaze information for human action prediction.

## 2 RELATED WORK

**Action prediction and activity forecasting tasks**    Action prediction and activity forecasting tasks both aim to predict future human actions. While action prediction tasks focus on predicting immedi-

ate actions given an input image sequence or video clip, activity forecasting emphasizes predicting broader human activities over a longer time frame (Gao et al., 2017). However, both the tasks share the same goal of understanding human behaviour and making reasonable predictions. These tasks have been extensively studied, particularly through the challenges established by the Ego4D benchmark (Grauman et al., 2022). Previous approaches have leveraged Long Short-Term Memory (LSTM) networks and modality attention mechanisms (Furnari & Farinella, 2019) to predict future actions. Other studies have combined bottom-up and top-down approaches to infer latent goals and predict actions by modeling temporal dynamics (Zhao et al., 2024a). Mascaro et al. (2024) decomposed the action prediction problem into predicting low-level actions derived from high-level intentions, which showed an improvement in prediction performance. Similarly,Ashutosh et al. (2023) presented a hierarchical approach for action anticipation by modeling both short-term low-level actions ("what the person is doing right now") and high-level long-term intentions ("what the person wants to do") using contrastive learning. Cho et al. (2024) highlighted the significance of interactions between the next active object and human hands for short-term anticipation, predicting the next active object first to model future interactions. Transformers and their variants have also been employed to model interactions between objects and hands, enhancing the prediction of future actions (Roy et al., 2024). In our work, we aim to predict future human actions in the form of fine-grained and descriptive annotations by incorporating eye gaze as an additional modality and employing a gaze-regularized attention mechanism.

**Attention and gaze-augmented models**   Attention-based models facilitate the discovery of important features and enhance performance on downstream tasks, and have been applied to various domains, including autonomous driving (Braunagel et al., 2017) , action prediction and human-computer interaction (Weber et al., 2020; Shafti et al., 2019; Aronson et al., 2018). For example - class activation maps have been used to leverage pooling layers in deep learning networks to generate class saliency maps (Sudhakaran & Lanz, 2018), allowing the model to focus on regions containing objects that correlate with the activity being considered for action anticipation and prediction. Guided-attention mechanisms have also been employed to model the next active object and predict future interactions (Thakur et al., 2023). The Spatiotemporal Attention Module (STAM) (Lu et al., 2019b) uses eye gaze information as supervision to train a network that predicts an attention map for activity recognition. Other studies similarly also use gaze as supervision to form attention maps for activity recognition tasks (Min & Corso, 2020; Awale & Sarikaya, 2022).

A lot of the previous work has utilized eye gaze data as ground truth to train attention prediction networks and to predict eye gaze. However, in our study, we use eye gaze data as a direct signal for human action prediction, building a gaze-regularized attention mechanism. With advancements in eye-tracking methods and devices, and the abundance of egocentric videos, eye gaze can be easily obtained and should be used as a direct signal. Our model conditions predictions on eye gaze data as an input signal to output fine-grained text annotations of human actions.

## 3 METHOD

We aim to develop a VLM that can accurately predict fine-grained human actions *in text* to facilitate human-machine interaction or collaboration in the physical world. In the following, we first formalize the problem and specify the training data, after which, we elaborate on the proposed mechanism that regularizes the attention in VLMs with gaze for enhanced predictions.

**Problem setup**   Given egocentric video frames (observation) $\{I_i\}_{i=1}^{\tau_o}$ over the past $\tau_o$ seconds (observation time), the *action prediction* VLM we aim to develop shall output text descriptions $\{\ell_i\}_{i=\tau_o+1}^{\tau_o+\tau_p}$ corresponding to future frames (what will happen) in the imminent $\tau_p$ seconds (prediction time). Let $\phi_{gaze}$ represent the gaze augmented VLM which aims to model the likelihood of the fine grained text descriptions (represented by $\ell_i$) when eye gaze information is also provided:

$$\phi_{gaze}(\ell, \{I\}, \{H\}, \{G\}) = \prod_{i=\tau_o+1}^{\tau_o+\tau_a} p(\ell_i \mid \ell_{<i}, I_{\leq\tau_o}, H_{\leq\tau_o}, G_{\leq\tau_o}), \qquad (1)$$

where $\{H\}$ represents the gaze heatmaps and $\{G\}$ represents the gaze overlaid images obtained by blending the gaze heatmaps and the video frames. It is shown by Lee et al. (2021) that human

brains make predictions of the future over a hierarchy of timescales: ranging from fine-grained changes of words or images (in a duration of 1 to 4 seconds) to coarse-grained evolution of movie plots (up to 15 seconds). Since we focus on developing a model that can anticipate detailed human activities to facilitate human-machine interaction, we set the prediction time '$\tau_p$' to 2 seconds by default. Furthermore, Seidel et al. (2014) discovered that attention control mechanisms, in response to duress or stimuli, are activated for up to 2 seconds before the actual event, thus the observation time '$\tau_o$' is set to a value larger than 2 seconds. An ablation study of different observation/prediction horizons is reported in Sec. 8.3.4 in the Appendix. Next, we detail the training data and how the gaze information is utilized to promote human action prediction.

## 3.1 DATASET CONSTRUCTION

We use the Ego4D dataset (Grauman et al., 2022), which includes video clips with eye-gaze data and textual action-object annotations. Gaze points are converted into images to highlight important visual regions. Videos are downsampled to one frame per second to reduce computational load.

For fine-grained action prediction, we enhance existing annotations with detailed textual descriptions using GPT-4V (OpenAI, 2023). This involves generating initial frame-by-frame descriptions and refining them by iteratively improving the prompt based on manual feedback. The final text annotations provide contextually coherent descriptions for each frame. During model training and testing, a sequence of images is input, and the subsequent text annotations serve as ground truth for action prediction.

Specifically, for every image frame $I_i$, we obtain a corresponding text annotation $l_i$. Figure 5 and Figure 6 in the Appendix provides a diagram illustrating this process, along with an example of the resulting annotations (Figure 7). For training and testing, the model is provided with a sequence of images $\{I_i\}_{i=1}^{\tau_o}$, while the text annotations $\{l_i\}_{i=\tau_o+1}^{\tau_o+\tau_p}$ corresponding to the image frames $\{I_i\}_{i=\tau_o+1}^{\tau_o+\tau_p}$ serve as the ground truth for predicting the anticipated actions. Here, $\tau_o$ represents the observation time, set to 5 seconds, and $\tau_p$ represents the prediction time, set to 2 seconds. Please refer to Sec. 8.1.1 for more details on the dataset construction pipeline. Next, we introduce the base model without eye gaze enhancement before delving into the proposed techniques that leverage eye gaze by regularizing the attention maps in the prediction.

## 3.2 BASE MODEL

Leveraging Flamingo's capability to model cross-modal interactions between images and language, we adapt the open-source Flamingo model (Awadalla et al., 2023) for our base model. We train the perceiver resampler in the vision block and the cross-attention layers in the language block, keeping other components frozen. The input image sequence $\{I_i\}_{i=1}^{\tau_o}$ is processed by a pre-trained ViT to extract visual features, which are then fed into a perceiver resampler, learning a fixed-size representation of the visual data and leveraging time embeddings to capture temporal relationships. The output from the perceiver resampler connects to the language module to generate predicted text annotations. This base model, trained without gaze information, serves as a benchmark for evaluating our gaze-augmented models. Let $\phi$ represent the base model, then the problem can be formulated as follows:

$$\phi(\ell, \{I\}) = \prod_{i=\tau_o+1}^{\tau_o+\tau_a} p(\ell_i \mid \ell_{<i}, I_{i \leq \tau_o}). \tag{2}$$

Furthermore, let $P(\ell)$ be the actual probability distribution of the ground-truth text, and $\phi(\ell, \{I\})$ the model predicted distribution. The cross-entropy loss to be minimized for training can be written as:

$$\mathcal{L}_{\text{CE}}(\{I\}, \ell) = -\sum P(\ell) \log(\phi(\ell, \{I\})). \tag{3}$$

This setup establishes a clear baseline, allowing us to measure the impact of incorporating gaze information in subsequent models after we elaborate on the regularization technique for enhanced human action prediction in the following.

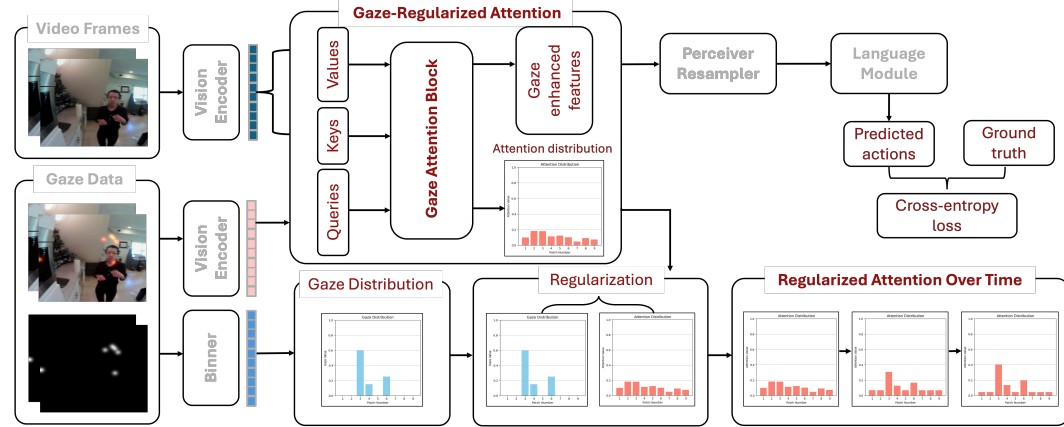

Figure 2: **Overview of the Architecture** A ViT encoder extracts features from video frames and gaze-overlaid images. A gaze-regularized attention block processes both the original and gaze-overlaid images, focusing on regions indicated by gaze patterns to produce gaze-enhanced features. These features are then passed to a Perceiver Resampler, which generates a fixed-size representation for the language module to predict future actions. A gaze regularizer aligns the model's attention with human gaze patterns by minimizing the Kullback–Leibler divergence between the attention distribution obtained from the attention block and the gaze distribution obtained from the binary heatmap images.

## 3.3 GAZE AUGMENTED MODEL

Human gaze patterns reveal key insights into attention and focus, providing clues about intentions, thoughts, and potential actions (Tipper, 2010; Frischen et al., 2007). Leveraging this natural attention mechanism, we propose a gaze-regularized attention block to enhance visual features from the pre-trained ViT before passing them to the Perceiver Resampler. The input to the gaze-regularized block includes a sequence of RGB images from the video, $\{I_i\}_{i=1}^{\tau_o}$, where $i$ denotes time in seconds and $\tau_o$ is the observation time. Additionally, we incorporate corresponding binary heatmap images $\{H_i\}_{i=1}^{\tau_o}$, highlighting regions of gaze. These heatmaps are generated from textual eye gaze data and are blacked out except for the gaze points, which are smoothed using a Gaussian filter to create a smooth heatmap. Moreover, these heatmaps generate a gaze distribution, indicating how gaze is distributed across image patches. Unlike the base model, the gaze-augmented model utilizes both eye gaze data and RGB images. The binary heatmap images and gaze-overlay images derived from the eye gaze data are shown in Figure 3. More information about the gaze-augmented model and it's components can be found in Sec.8.2 of the Appendix.

### 3.3.1 GAZE DISTRIBUTION COMPUTATION

Each binary heatmap image is first divided into patches. Since gaze is usually concentrated on specific regions, not every patch will contain gaze information. Pixels occupied by gaze are indicated by non-zero pixel values, while pixels without gaze are blacked out, having values of zero. For a binary heatmap image denoted as $H_t$, let $N$ represent the total number of patches. The proportion of gaze occupied within a patch $N_{i,j}$, where $(i,j)$ represents the patch's position in the grid, can be calculated as follows:

$$N_{i,j} = \frac{\sum_{y=\frac{j(h)}{n_v}}^{\frac{(j+1)(h)}{n_v}} \sum_{x=\frac{i(w)}{n_h}}^{\frac{(i+1)(w)}{n_h}} p_{xy}}{\sum_{y=0}^{h} \sum_{x=0}^{w} p_{xy}} \quad \text{for } i \in \{0, n_h - 1\} \text{ and } j \in \{0, n_v - 1\}. \tag{4}$$

Here, $h$ and $w$ represent the height and width of the image, $n_h$ and $n_v$ denote the number of horizontal and vertical patches, respectively, and $p_{xy}$ is a binary variable representing the pixel value at position $(x, y)$, which can be either 0 or 1. The denominator corresponds to the total value of all pixels in the image. For each binary heatmap image $H_t$, we derive a gaze distribution $\hat{H}_t$, which is a vector of shape $(1, n_h \times n_v)$. The number of patches, $N$, in the gaze distribution is equal to

the number of tokens produced per image by the ViT. This gaze distribution plays a crucial role in the gaze regularizer used in the attention block, acting as the target distribution that we want the attention distribution to mimic, which will be discussed in the following section.

### 3.3.2 Gaze-regularized attention block

To obtain features from the RGB images and gaze-overlaid images respectively, we utilize the ViT, and additional information can be found in Sec. 8.2.1 of the Appendix. The features derived from the gaze-overlaid images serve as queries ($Q$) in the gaze-regularized attention block. Simultaneously, the scene image features from the RGB images and are used as keys ($K$) and values ($V$). Attention is then calculated based on the following equation:

$$\text{Attention}(Q, K, V) = \text{softmax}\left(\frac{QK^T}{\sqrt{d_k}}\right) V = AV, \tag{5}$$

where $A$ represents the attention weights. Simply using gaze features as queries does not guarantee that the attention will be concentrated on gaze-allocated regions. The attention scores will be distributed across the image. We aim to guide the model to prioritize these regions, directing more focus (attention) towards areas where gaze is present. To this end, we introduce a gaze regularizer designed to enhance the model's attention on gaze-allocated regions.

The gaze regularizer is implemented using Kullback-Leibler (KL) divergence, which is applied to the training of the gaze-augmented model. The aim of the regularizer is to ensure that the distribution of the attention weights is more closely aligned with the gaze distribution obtained from the binary heatmap image. We denote the attention weight distribution as $A$, while the gaze distribution is represented by $H$. The KL divergence for the two distributions can be obtained using the following equation:

$$D_{\text{KL}}(A\|H) = \sum_i A_i \log \frac{A_i}{H_i}. \tag{6}$$

The binary heatmap images are not used as queries because doing so could lead to important image information being overlooked. For instance, the heatmap images for two totally different images could be the same if one looks at the center of the image.

To avoid this, we use gaze-overlaid images as queries, which preserve the gaze information while also retaining the unique visual content of each image. It is crucial to preserve the visual content while aligning the attention distribution with the gaze distribution to ensure the model properly integrates both sources of information.

The overall objective function we minimize to train the gaze-augmented model is as follows:

$$\mathcal{L}_{\text{total}} = \mathcal{L}_{\text{CE}} + \lambda * D_{\text{KL}}(A\|H), \tag{7}$$

where $\lambda$ is the coefficient of the regularization term in the proposed objective function, and the cross-entropy loss is similar to the loss from the base model. By utilizing this gaze-regularized attention mechanism within the model, we can orient the model's attention to regions highlighted by gaze, and thus improve the quality of the predicted future actions.

### 3.3.3 Singular and Aggregated gaze augmented models

In eye-tracking literature, eye movements are primarily categorized into fixations and saccades. During a fixation, the eyes pause and focus on a specific area, gathering detailed visual information. Most visual data is acquired during fixations, which typically last around 200 ms (Rayner, 2009). Although small involuntary eye movements may occur during fixations, they still concentrate on a specific region of interest. Saccades, on the other hand, are rapid eye movements that shift focus from one point of interest to another, causing significant changes in the visual scene. Saccades are crucial for visual search, and two consecutive saccades can cover considerable distances. Given the nature of saccades and fixations, as well as their average durations, detailed or relevant information is typically collected within a temporal window of $\delta = 200$ milliseconds.

Correspondingly, we provide both a singular gaze-augmented model and an aggregated gaze-augmented model. For each image frame $I_t$, we generate a heatmap image $H_t$ and a gaze-overlaid image $G_t$ using gaze coordinates available at time $t$. Gaze data from the Ego4D dataset (Grauman

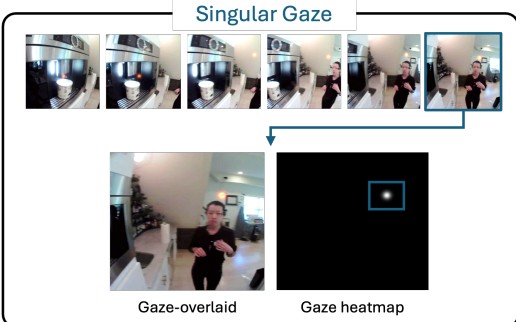

Figure 3: **Heatmap creation.** Illustration of gaze data collection for generating gray-scale heatmaps and gaze-overlaid images. On the left, the aggregated gaze model incorporates multiple gaze points collected over the interval $[t - \delta, t]$ to generate the heatmap. On the right, the singular gaze model uses a single gaze point collected at time $t$. Both with Gaussian smoothing.

et al., 2022) is sampled at 30 frames per second and stored in the format $g_t = (\text{timestamp}, x, y)$, where $(x, y)$ is the coordinate of the gaze point. For the singular gaze-augmented model, the heatmap at time $t$, $H_t$, is:

$$H_t = f(g_t). \tag{8}$$

Here, the function $f$ represents a mechanism to construct the heatmap, such as a Gaussian smoothing. In the singular gaze-augmented model, we use a single gaze point captured at the timestamp nearest to when the RGB image is obtained.

However, a singular gaze point can be noisy due to measurement errors or micro-saccades (Rolfs, 2009; Ratliff & Riggs, 1950; Collewijn & Kowler, 2008). An aggregated gaze pattern can account for such variations. To minimize the impact of noise and ensure a sufficient time frame for collecting detailed information, we propose aggregating points within a fixed time interval $\delta$ around the image frame $I_t$. The corresponding heatmap $H_t$ at time $t$ is then:

$$H_t = \sum f(g_i), \quad \forall i \in [t - \delta, t], \tag{9}$$

which is normalized by the length of the interval, and $g_i$ for all $i \in [t - \delta, t]$ represents the set of gaze points lying within the vicinity of timestamp $t$. These gaze points are used to obtain both the binary heatmap image $H_t$ and the gaze-overlaid image $G_t$. However, aggregation can sometimes include gaze points that may be occluded in the final frame. To address this, we implement an occlusion check using forward and backward optical flow consistency. The purpose of the occlusion check is to ensure that if significant occlusion occurs, the corresponding gaze points are excluded from the heatmap creation process. More information about the occlusion check can be found in Sec. 8.2.3. An example of using the occlusion check is shown in Figure 10.

To recap, for the aggregated gaze-augmented model, we collect gaze data points over a fixed interval, denoted as $\delta$, to establish a more informative gaze pattern. This aggregation of gaze points, after an occlusion check, is then used to create a heatmap, which is subsequently overlaid on the RGB scene image to produce the gaze-overlaid image. Next, we demonstrate the effectiveness of the proposed gaze-regularization mechanisms for human action prediction.

## 4 EXPERIMENTS

In this section, we begin by outlining the evaluation metrics used to assess model performance. Then, we compare the base models with the gaze-augmented models to emphasize the significance of incorporating gaze as a signal along with gaze-regularized attention. Following this, we explore multiple variations of the gaze-augmented models to identify the best-performing configuration and to justify the effectiveness of gaze in different settings discussed in the previous section. Finally, we present evidence supporting the use of finer-grained text annotations for gaze-augmented models to enhance overall performance. Additional information about the other experiments conducted as part of our study can be found in Sec.8.3 whereas the training and evaluation details for the models can be found in Sec.8.5.

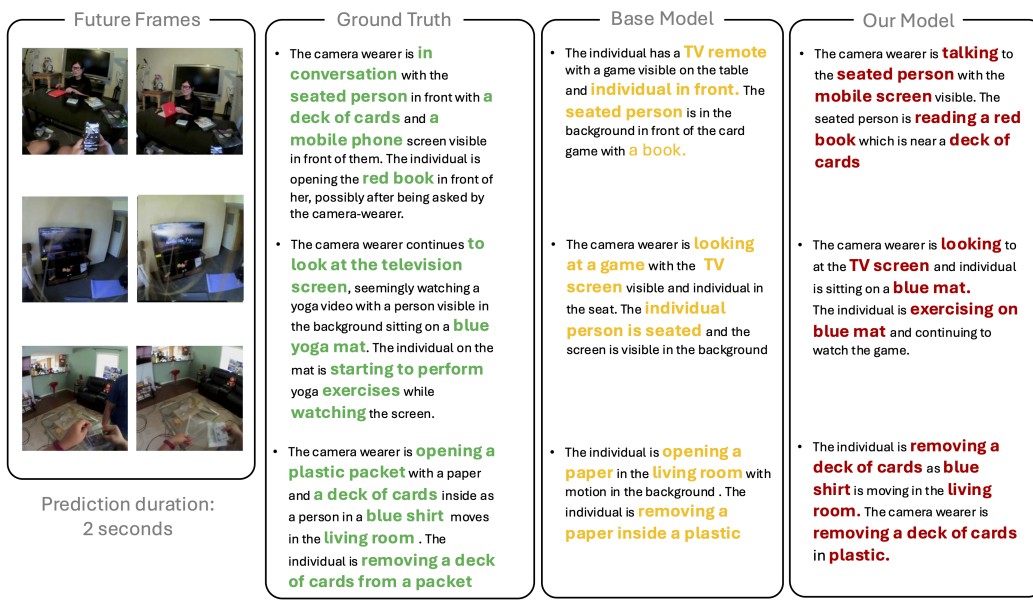

Figure 4: Action prediction results for the base model (without gaze) and our gaze-augmented model are presented for an observation horizon of 5 seconds. Past frames are omitted, but ground-truth annotations and future frames with a prediction duration of 2 seconds are provided as references. Keywords from each set of annotations are highlighted for easier reading.

## 4.1 EVALUATION METRICS

For evaluation, we propose using a semantic transformer (Reimers & Gurevych, 2019) to provide a quantifiable score that compares the generated output with the ground-truth action text. This scoring system is designed to ensure that the model does not penalize variations of sentences that are semantically similar. Additionally, we aim for the model to penalize sentences where the word order is nonsensical or incomprehensible to humans. Alongside the semantic score, we also utilize the ROUGE-L score and METEOR score, which are widely employed to assess text similarity.

## 4.2 COMPARISON BETWEEN BASE MODEL AND GAZE-AUGMENTED MODELS

The objective of this experiment is to quantify the impact of eye gaze data on human action prediction by comparing model performance using measurable metrics. The base model receives only RGB image frames as input, while the gaze-augmented models incorporate multiple variations of gaze information. All gaze-augmented models share the same architecture but differ in the way gaze data is utilized. In the Singular Gaze-Augmented Model, for each image frame $I_t$ at time $t$, the corresponding heatmap $H_t$ is generated using a single gaze point $g_t$, obtained from the timestamp closest to $t$. In the Aggregated Gaze-Augmented Model, for each image frame $I_t$ at time $t$, the corresponding heatmap $H_t$ is generated using aggregated gaze points $g_k$, where $k \in [t - \delta, t]$ and $\delta$ is set to 200 milliseconds.

All models are trained under identical conditions. As shown in Table 1, the gaze-augmented models outperform the base model across all evaluation metrics. Using aggregated gaze points over a 200-millisecond interval shows a noticeable improvement in performance compared to using a single gaze point. This can be attributed to the larger focus area and the temporal gaze pattern enabled by the aggregation, allowing the model to capture a broader context for inferring the intent and better facilitate the anticipation of future actions. Please refer to Figure 4 for qualitative results.

## 4.3 EFFECT OF THE GAZE REGULARIZER

We assess the impact of the gaze regularizer using the aggregated gaze-augmented model to determine its importance. The final objective function includes the gaze regularization coefficient $\lambda$, and

Table 1: Evaluation of base model and gaze-augmented models with varying granularity in action annotations (i.e., GPT-4V and ShareCaptioner).

| Model | Annotation source | Semantic Score($\uparrow$) | Meteor Score($\uparrow$) | Rouge-L ($\uparrow$) | | |
|---|---|---|---|---|---|---|
| | | | | Precision | Recall | F-score |
| Base (no gaze) | GPT-4V | 0.6525 | 0.4075 | 0.4335 | 0.4301 | 0.4318 |
| Singular Gaze | GPT-4V | 0.7316 | 0.4501 | 0.4822 | 0.5309 | 0.5054 |
| **Aggregated-Gaze** | GPT-4V | **0.7826** | **0.5033** | **0.5193** | **0.5644** | **0.5405** |
| Base (no gaze) | ShareCaptioner | 0.6437 | 0.5060 | 0.5730 | 0.5566 | 0.5646 |
| Singular Gaze | ShareCaptioner | 0.8212 | 0.6514 | 0.6906 | 0.6914 | 0.6905 |
| **Aggregated-Gaze** | ShareCaptioner | **0.9125** | **0.7114** | **0.7617** | **0.7717** | **0.7666** |

Table 2: Effect of regularization on aggregated gaze-augmented models.

| Regularization coefficient | Semantic Score($\uparrow$) | Meteor Score($\uparrow$) | Rouge-L ($\uparrow$) | | |
|---|---|---|---|---|---|
| | | | Precision | Recall | F-score |
| 0 | 0.6317 | 0.4094 | 0.3872 | 0.3622 | 0.3738 |
| **100** | **0.7826** | **0.5033** | **0.5193** | **0.5644** | **0.5405** |
| 1000 | 0.7798 | 0.4963 | 0.5127 | 0.5558 | 0.5330 |

we evaluate the model's performance on the test set using different values of $\lambda$. When $\lambda = 0$, the gaze regularizer is inactive and has no effect on the loss function, while larger values of $\lambda$ increase its influence. As shown in Table 2, the performance of the aggregated gaze-augmented model without regularization is comparable to the base model. Optimal performance is achieved at moderate $\lambda$ values, with a slight decline at higher regularization levels. The diminished performance without regularization underscores the significance of aligning attention with the gaze distribution through the gaze regularizer when the gaze attention block is included.

## 4.4 GAZE-REGULARIZED ATTENTION BLOCK

The gaze-regularized attention block is a critical addition to the baseline OpenFlamingo model. In our ablation studies, we examine the model's performance as the number of gaze-regularized attention blocks varies, aiming to identify the optimal number of blocks for further training. As shown in Table 3, the performance improves up to $n = 2$, after which it slightly declines with additional blocks. Further training confirms that using two gaze-regularized attention blocks yields the best results, as this configuration balances aligning the gaze and attention distributions while preventing over-alignment.

## 4.5 EFFECT OF OCCLUSION CHECK ON AGGREGATED GAZE-AUGMENTED MODELS

In the aggregated gaze model, gaze points are collected within a specified time interval $\delta$ (200 ms). However, in dynamic environments, aggregating gaze points from the interval $[t-\delta, t]$ may introduce inaccuracies due to changes in the scene or camera movement. To mitigate this, we introduced an

Table 3: Effect of the number of gaze-regularized attention blocks on the performance of the aggregated gaze-augmented models.

| Attention blocks | Semantic Score($\uparrow$) | Meteor Score($\uparrow$) | Rouge-L ($\uparrow$) | | |
|---|---|---|---|---|---|
| | | | Precision | Recall | F-score |
| 1 | 0.7434 | 0.4945 | 0.5065 | 0.5630 | 0.5328 |
| **2** | **0.7826** | **0.5033** | **0.5193** | **0.5644** | **0.5405** |
| 5 | 0.7765 | 0.5013 | 0.5098 | 0.5528 | 0.5301 |

Table 4: Comparison of the aggregated gaze-augmented models with and without occlusion check.

| Aggregated gaze | Semantic | Meteor | Rouge-L (↑) | | |
|---|---|---|---|---|---|
| model | Score(↑) | Score(↑) | Precision | Recall | F-score |
| w/o occlusion-check | 0.7616 | 0.4718 | 0.5090 | 0.5508 | 0.5286 |
| **with occlusion-check** | **0.7826** | **0.5033** | **0.5193** | **0.5644** | **0.5405** |

occlusion check to ensure that only relevant gaze points are aggregated. More details about the occlusion check method can be found in Sec. 8.2.3 in the Appendix.

To evaluate the impact of this adjustment, we conducted experiments comparing models with and without the occlusion check. As shown in Table 4, the model incorporating the occlusion check slightly outperforms the one without it. The difference in the evaluation metrics can be attributed to the fact that only relevant and accurate gaze points are considered, which reduces noise and prevents the model from being confused by irrelevant data.

### 4.6 IMPACT OF THE ANNOTATION QUALITY ON MODEL PERFORMANCE

To facilitate our experiments, we designed a prompt-based method to automatically generate text annotations for image frames using GPT-4V. Although these annotations were more detailed than the originals, we also generated additional annotations using ShareCaptioner (Chen et al., 2024), a tool specifically designed to produce comprehensive captions that capture changes between frames. We aim to compare the descriptive quality of these annotations and evaluate their impact on both the base and gaze-augmented models.

We find that annotations generated by ShareCaptioner are notably more detailed and fine-grained than those from our initial setup. While the base model's performance remained similar with both sets of annotations, the gaze-augmented model showed a nearly 12 percent improvement when using the more fine-grained annotations, as shown in Table 1. This improvement can likely be attributed to the model's ability to leverage areas of concentrated gaze when processing more descriptive captions, allowing for better prediction of detailed actions. From the experiment, we conclude that the importance of eye gaze increases even further when the granularity of the annotations is higher.

## 5 CONCLUSION

Our work demonstrates that incorporating gaze data significantly enhances the predictive abilities of VLMs, proving valuable for applications requiring precise human action prediction, such as assistive robotics and human-machine collaboration. By integrating eye gaze data, our approach improves the model's focus on critical image features, leading to enhanced prediction accuracy and reliability.

Our gaze-augmented model is trained with a gaze-regularized attention strategy, which simulates the natural attention mechanism driven by human gaze. This integration results in improved semantic and ROUGE scores, underscoring the effectiveness of gaze-regularized attention in boosting model performance. Through additional experiments, we highlight the crucial role of employing a gaze regularizer and gaze-augmented attention blocks, which provide insight into how to maximize the utilization of gaze information.

Looking ahead, we aim to further refine our model by enhancing the integration of gaze data and exploring other training techniques. We also plan to develop a dedicated large-scale dataset to support other researchers in running their models and advancing this methodology. To facilitate further research and development, we will make our code and dataset publicly available, allowing others to build upon our architecture and enhance its capabilities.

## 6 ETHICS STATEMENT

We declare that our research does not present any potential ethical issues and is in compliance with the ICLR code of ethics. The study does not involve human subjects, sensitive data, or methodolo-

gies that could result in harmful outcomes or biases. All data this work uses is publicly available, and no privacy or security concerns are implicated.

## 7 REPRODUCIBILITY STATEMENT

We have made significant efforts to ensure the reproducibility of our work. A code example is provided as supplementary material, demonstrating the core components of our approach. Upon acceptance, we will release all of the data and the complete training and testing code to facilitate the full reproducibility of our results.

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

# 8 APPENDIX

In this appendix, we present supplementary material related to our study. This includes detailed information on the dataset creation process and the design of the prompts used in our experiments. We also include results from several ablation studies conducted during our research. Finally, we provide essential details about the model training process to assist readers who may wish to reproduce our work from scratch.

## 8.1 DATASET

The Ego4D and Epic-Kitchens datasets consist of egocentric videos capturing camera-wearers performing daily activities in controlled settings (Grauman et al., 2022; Damen et al., 2022). Other relevant datasets include the EGTEA+ Gaze dataset, which offers 28 hours of content focused on cooking and kitchen activities (Li et al., 2020), and the more recent Visual Data Experience (VDE) dataset, which contains around 240 hours of clips documenting day-to-day activities coupled with gaze and head tracking (Greene et al., 2024).

These datasets are supplemented with text annotations in the form of (action, verb) tuples as well as coarse-grained narrations, which are used for action anticipation and activity forecasting challenges. For our study, we collected visual eye gaze data from the Ego4D dataset and modified the text annotations from coarse-grained descriptions to more descriptive fine-grained descriptions. In the future, we plan to incorporate the more recent VDE dataset into our study as well.

In this section, we provide details about the dataset creation process used for model training and testing, the prompt design, and alternative sources of annotations in the form of VLMs.

### 8.1.1 DATASET CONSTRUCTION AND PROMPT DESIGN

The Ego4D dataset comprises egocentric video clips along with supplementary data such as audio, text annotations, eye gaze data, and additional metadata (Grauman et al., 2022). For our project, we focused on the subset of video clips that include eye gaze data, containing approximately 33.3 hours of egocentric videos recorded from 80 participants. The eye gaze data is provided in numerical form, containing canonical timestamps and the pixel coordinates of gaze points. We transform gaze points to images to represent important visual regions, aligning with how humans perceive spatial information (Laeng et al., 2014). Due to the minute differences between consecutive images in the original videos, we perform downsampling to one image per second to reduce computational requirements while remaining effective.

Since our focus is fine-grained action prediction from ego-centric videos, we augment existing annotations with detailed textual descriptions to enhance human-machine interaction. We leverage GPT-4V to generate annotations for video frames by processing a sequence of images and prompting it to describe each frame. This method ensures contextual coherence across frames. After obtaining initial descriptions, we manually evaluate their quality and provide feedback to refine the output. This feedback is used to modify the prompt, which is then fed back into GPT-4V with the image frames. We conduct prompt selection and refinement using multiple sequences from different video clips to ensure generalizability. This iterative process continues until we establish an optimal prompt template that consistently yields accurate and contextually appropriate annotations, confirmed through human evaluation.

Specifically, we start by passing a small set of images to GPT-4V with a basic prompt like: "Describe what is happening in the image sequence and output the text descriptions." The initial output was manually evaluated, and feedback was provided. This feedback, along with the original prompt, was passed to ChatGPT to refine the prompt for generating accurate and meaningful text annotations. The manual evaluation process ensured that the generated annotations met the following criteria:

1. A clear description of the objects being manipulated or focused on in the scene.

2. A detailed account of the actions being performed by both the camera wearer and other individuals present in the images.

3. Information about any trajectories or movements that take place within the scene.

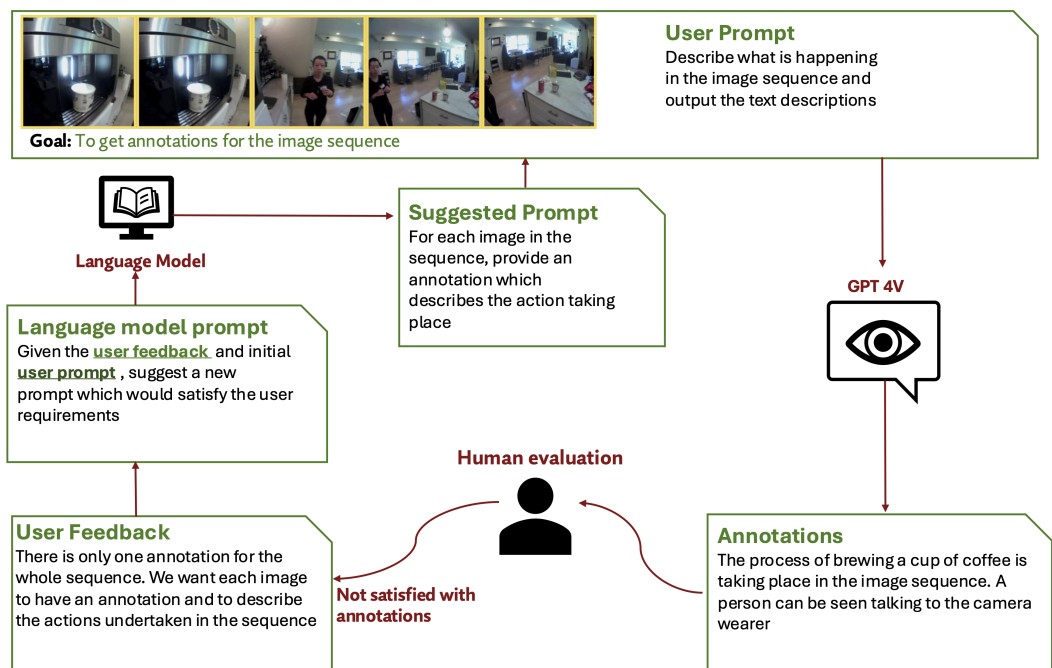

Figure 5: **GPT-4V Iterative Prompting Workflow.** The process begins with a sequence of images and an initial prompt, which are input into GPT-4V to generate annotations for each image. The user then evaluates these annotations and provides feedback, which is incorporated into the prompt using a language model. This modified prompt is used to refine the annotations in a continuous cycle. The objective is to improve the quality and relevance of the output with each iteration until the user is satisfied. For GPT-4V, a set of 10 images is provided at once, ensuring that the annotations maintain contextual coherence.

4. Clear, fine-grained annotations that fulfill these criteria in a way that is easily understandable by both humans and machines, such as robots that may use these instructions for task execution.

After several iterations of refining the prompt and evaluating the results, we identified a suitable template for generating high-quality text annotations. This final template was then used to annotate the image sequences using GPT-4V. More details on the prompt design process can be found in Figure 5 and Figure 6.

### 8.1.2 TEXT ANNOTATIONS USING AN ALTERNATE VLM

In the previous section, we obtained text annotations using GPT-4V. In addition, we conduct further experiments using an alternate VLM to generate text annotations for the same video clips.

Specifically, we employed ShareCaptioner (Chen et al., 2024) to annotate the entire dataset. Under similar training conditions, we compared the empirical performance of the models using annotations from GPT-4V and ShareCaptioner. The purpose of this comparison was to assess whether there is a significant performance variation – either a drop or improvement – when text annotations are sourced from a different VLM. Additionally, we aimed to confirm whether the trend of superior performance in gaze-incorporated models persists regardless of the annotation source. Finally, we sought to investigate if the granularity of the annotations impacts the model's performance.

As part of this investigation, we also experimented with an open-source implementation of LLaVA to generate finer-grained annotations Lin & Long (2024). However, upon manual evaluation, we found no significant difference between the annotations produced by ShareCaptioner and those generated by LLaVA. In order to have a dedicated comparison between fine-grained annotations and finer-grained annotations on the base model and the gaze-augmented model, we decided to proceed

with GPT-4V annotations and ShareCaptioner annotations. However, a comparison of the semantic scores obtained by the base model and aggregated gaze-augmented models for all three annotation models can be found in Figure 7. We observed that the gaze-augmented models exhibited a 10-12 percent improvement in performance when utilizing the finer-grained annotations, indicating that more detailed descriptions can further enhance model accuracy in predicting future actions.

Examples of the annotations obtained for a sample image, as well as the semantic scores for the base models and the gaze-augmented models, can be seen in Figure 7. These examples highlight the difference in performance under varying quality of the fine-grained text annotations.

**Note:** It is important to emphasize that our goal is not to compare the performance of ShareCaptioner and GPT-4V, but rather to evaluate the quality of the annotations obtained from each system and the effect of the quality on the gains of employing gaze in the prediction. The differences in annotation granularity may be influenced by cost and time constraints, as the setups used to generate these annotations varied. Therefore, a direct comparison of the VLMs themselves would be inappropriate.

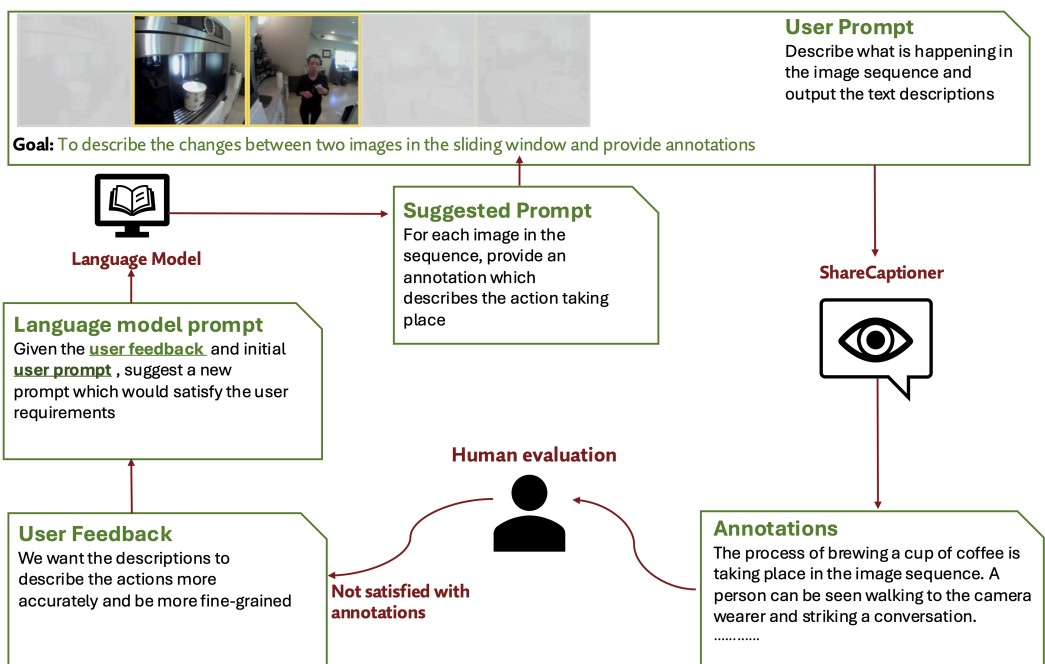

Figure 6: **ShareCaptioner Iterative Prompting Workflow.** The process begins with a sequence of images accompanied by a starter prompt, which is processed by ShareCaptioner to generate initial annotations. The user then reviews these annotations and provides feedback. This feedback is integrated into the prompt using a language model. The updated prompt is subsequently used to enhance the annotations through an iterative cycle. This cycle continues until the output meets the user's satisfaction, focusing on improving both quality and relevance. Additionally, a sliding window mechanism is employed to traverse the image sequence, capturing and describing changes as they occur to ensure that all actions are accurately recorded.

## 8.2 MODEL OVERVIEW AND COMPONENTS

In our study, we employ the open-source version of the Flamingo model (Awadalla et al., 2023), built upon the foundational work of the original Flamingo developed by Alayrac et al. (2022). The Flamingo is a VLM designed to leverage interleaved text and image data. It features a pre-trained vision encoder to extract input features, a trainable perceiver resampler for creating fixed-size representations of the input features, and trainable cross-attention layers on the language side. The latent features learned by the perceiver resampler are integrated into the language module. These features, along with the processed text, are input to the cross-attention layers in the language module which combines the visual features with the language features. The model utilizes a cross-entropy

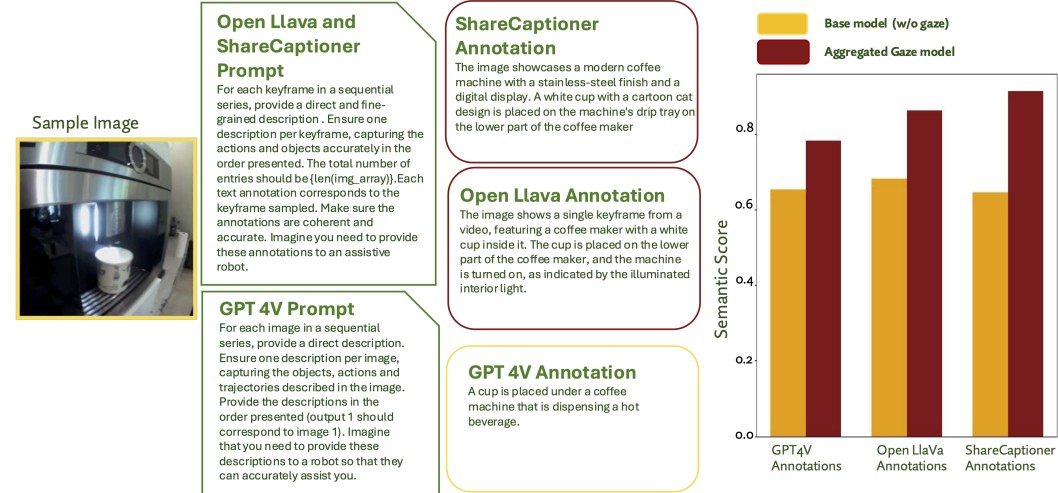

Figure 7: **Left**: Examples of annotations generated for a sample image by three different annotators: ShareCaptioner, Open Llava, and GPT-4V. The annotations from ShareCaptioner and Open Llava are more fine-grained compared to those produced by GPT-4V in our experimental setup. **Right**: A graph comparing the performance of the base model and gaze-augmented models, illustrating the impact of using text annotations of varying granularity.

loss mechanism, aiming to maximize the probability of predicting the correct text token based on preceding image and text.

In addition to traditional input modalities, such as RGB images, recent advancements have highlighted the potential of integrating various modalities beyond vision, including gaze, gait, and tactile sensors (Boshoff et al., 2024; Yang et al., 2024). Building on this premise, our approach incorporates eye gaze as an additional signal. Specifically, we introduce a gaze-regularized attention mechanism to make use of eye gaze as a signal. The base model without eye gaze data serves as a benchmark and relies exclusively on RGB images for input. In contrast, the gaze-augmented model integrates eye gaze data alongside the RGB images. The key distinction of the gaze-regularized attention models is their ability to condition the output using both RGB images and eye gaze data.

In the following section, we provide more details about some of the components in the model architecture, as well as provide information about the occlusion check which is used for gaze point correction during aggregation of gaze points.

### 8.2.1 VISION TRANSFORMER ENCODER

In both the base and gaze-augmented models, we utilize a pre-trained Vision Transformer (ViT) as the image encoder. Images are processed as sequences, where each image is tokenized into patches that are flattened and then transformed into embeddings. To maintain spatial relationships between the patches, these embeddings are supplemented with positional embeddings, ensuring coherence across the entire image. An illustration of the ViT architecture is provided in Figure 8. In the figure, as an example, we show how an image is divided into nine tokens, each of which is assigned a positional embedding. These tokenized patches are then passed through the pre-trained transformer encoder, resulting in an embedding that represents the original sequence of image patches. The pre-trained vision encoder employed in our study is ViT-L-14, which results in the formation of 256 tokens for each image. In the example shown in Figure 8, the observation length ($\tau_o$) is set to 5 seconds, meaning that the image sequences span a 5-second interval. The primary distinction between the original design and the approach used in our study is the omission of the final MLP head, as our focus is exclusively on extracting image features.

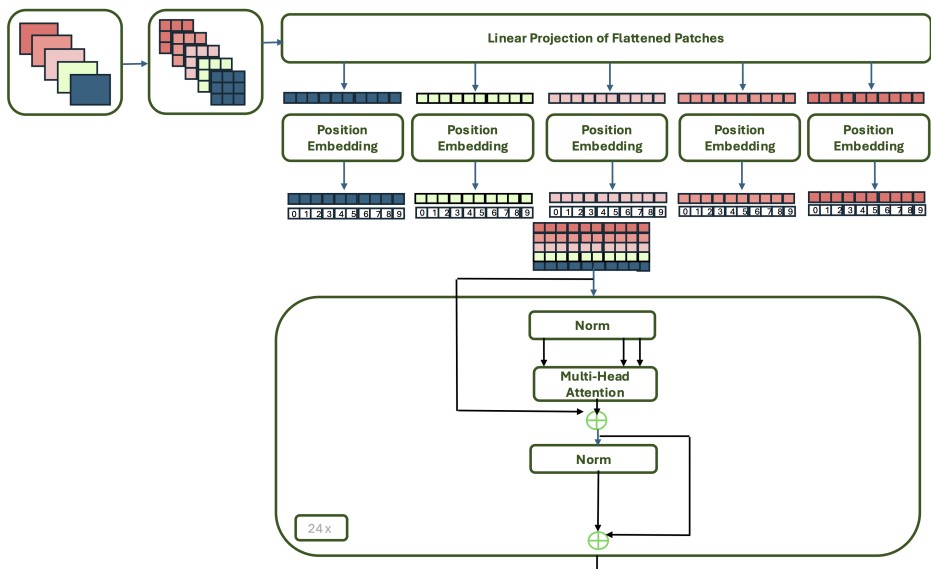

Figure 8: An example of how images are processed by the Vision Transformer in our model. In this case, each image is tokenized into 9 patches. Positional embeddings are then added to the patches, which are subsequently passed through an attention module. In this model, the final MLP head is omitted, as we are focused solely on extracting image features for further analysis.

### 8.2.2 GAZE-GUIDED ATTENTION BLOCK

In our study, we propose the inclusion of a gaze-regularized attention mechanism, which features a gaze-guided attention block that operates in conjunction with the gaze regularizer. In the gaze-guided attention block, gaze-overlaid images are processed through the ViT to extract features that serve as queries for the attention module. These queries contain information about both the scene and the regions occupied by gaze, leading us to refer to this block as the gaze-guided attention block. Once the gaze-based features are obtained, they pass through a learnable linear layer to produce the queries involved in the attention mechanism. The visual features obtained from the RGB image frames, after passing through the ViT, are used as keys and values within the gaze-guided attention block. Similar to the gaze-based features , the features obtained from the RGB image frames pass through a linear layer to form the keys and values for the attention mechanism. The final output, consists of gaze-enhanced features that are informed by the gaze-based queries.

Additionally, the attention distribution is calculated, reflecting how attention is allocated across all tokenized patches. Initially, this attention distribution may not align with the gaze distribution. Therefore, through regularization, we aim to progressively align the attention distribution with the gaze distribution over time. A closer look at the gaze-guided attention block can be found in Figure 9.

### 8.2.3 GAZE AGGREGATION WITH OCCLUSION CORRECTION

The input image sequence can exhibit dynamic changes due to environmental movement or the movement of the camera wearer. The method for aggregating gaze points is suitable only when the frames within the $\delta$ interval for which aggregation is done, shows moderate movement. However, preventing movement in a dynamic environment is challenging. If we aggregate gaze points in the $[t - \delta, t]$ interval to construct the heatmap $H_t$ and there is major occlusion between the earlier frames $[t - \delta, t)$ and the final frame at timestamp $t$, it becomes impractical to include gaze points from the occluded frames. In such cases, collecting gaze points from occluded frames may lead to inaccurate or misleading representations in the heatmap. To alleviate this issue, we perform an occlusion check between each frame in the $[t - \delta, t)$ interval and the final frame at time $t$ to ensure appropriate gaze aggregation. In the case of significant occlusion or a drastic change in the scene,

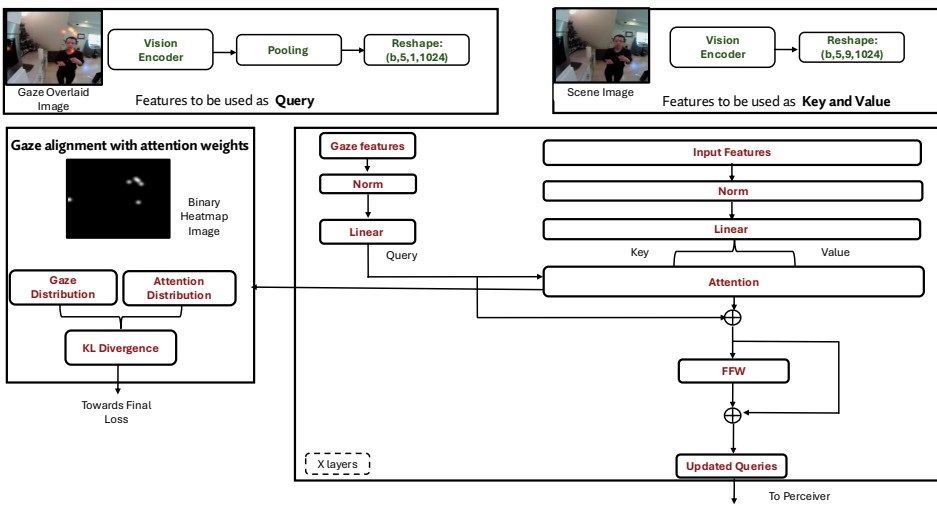

Figure 9: Closer look at the gaze-guided attention block in the pipeline. Gaze overlaid images are passed as queries, whereas the corresponding RGB images are passed as key and value. The binary heatmap image is used to obtain the gaze distribution, which is used in the regulariser by comparing it with the attention distribution obtained from the gaze-guided attention block. The regulariser attempts to align the attention distribution more towards the gaze distribution.

gaze points corresponding to the earlier frames should not be collected for the aggregation and subsequent formation of the heatmap $H_t$.

Using a method similar to the consistency check with optical flow presented by Hur & Roth (2017), we explicitly exclude gaze points that are occluded in the current frame. If a pixel is correctly translated and there is no major occlusion, then the difference between the forward optical flow displacement of this pixel $(x, y)$ and the displacement of the translated pixel $(\hat{x}, \hat{y})$ with backward optical flow should be close to zero.

For an RGB image $I_t$ at time $t$, we gather the image frames $\{I_k\}$ for all $k \in [t-\delta, t)$. Let the forward optical flow between images $\{I_k\}$ and $\{I_t\}$ in the horizontal direction be denoted by $Fx_{k \to t}$ and the backward optical flow by $Fx_{t \to k}$. Similarly, $Fy_{k \to t}$ represents the forward optical flow in the vertical direction, and $Fy_{t \to k}$ represents the backward optical flow in the vertical direction. Let the coordinates of a designated pixel $p_i$ be $(x_i, y_i)$. The new coordinates of the translated pixel in the subsequent frame, using optical flow, are computed as follows:

$$
\begin{aligned}
\hat{x}_i &= x_i + Fx_{k \to t}(x_i), \\
\hat{y}_i &= y_i + Fy_{k \to t}(y_i).
\end{aligned}
\tag{10}
$$

Next, we calculate the distance moved by this designated pixel in the horizontal and vertical directions according to the following equations:

$$
\begin{aligned}
d_{x_i} &= |Fx_{k \to t}(x_i)| - |Fx_{t \to k}(\hat{x}_i)|, \\
d_{y_i} &= |Fy_{k \to t}(y_i)| - |Fy_{t \to k}(\hat{y}_i)|.
\end{aligned}
\tag{11}
$$

If the observed proportion of pixels $\eta_{\text{observed}}$ exceeding the distance discrepancy is more than a predefined threshold $\eta$, we conclude that a major occlusion has occurred; otherwise, the occlusion is minor. The observed proportion of such pixels $\eta_{\text{observed}}$ is calculated as:

$$
\eta_{\text{observed}} = \frac{\sum_{i=1}^{h \times w} \mathbf{1}_{\text{condition}} \left( \sqrt{(d_{x_i})^2 + (d_{y_i})^2} > \epsilon \right)}{\sum_{i=1}^{h \times w} 1},
\tag{12}
$$

where the denominator represents the total number of pixels in the image.

We disregard the gaze points for frames $\{I_k\}$ where there is major occlusion with respect to the image frame $I_t$. If the occlusion is minor, the appropriate gaze points $\{g_i\}$ for all $i \in [t - \delta, t]$ are then translated into their new coordinates $\{\hat{g}_i\}$ and collected for the formation of heatmap $H_t$ and subsequently, for the gaze-overlaid image $G_t$. The transformed gaze points are computed as follows:

$$(\hat{g}_i)_x = (g_i)_x + Fx_{k \to t}((g_i)_x),$$
$$(\hat{g}_i)_y = (g_i)_y + Fy_{k \to t}((g_i)_y). \tag{13}$$

As mentioned above, the idea is that if there is a major occlusion, the difference between the distance traversed by a pixel during forward optical flow, and the distance traversed by the translated pixel during backward optical flow will be significantly greater than in cases where the occlusion is minor. Optical flow was calculated using the implemntation of the RAFT model developed by Teed & Deng (2020). Human feedback was utilized to distinguish between major and minor occlusions on sample image sequences, which informed the selection of hyperparameters $\epsilon$ and $\eta$. The hyperparameter $\epsilon$ is the threshold distance, which was set to 20, whereas $\eta$ is the threshold proportion of pixels that have exceeded the occlusion limit, set to 0.60. An example of the result of our occlusion check can be found in Figure 10. It can be observed that if the occlusion check is not present, the aggregation points do not accurately reflect where the person was looking in the final frame on which gaze is overlaid. The occlusion check ensures that the correction is effective.

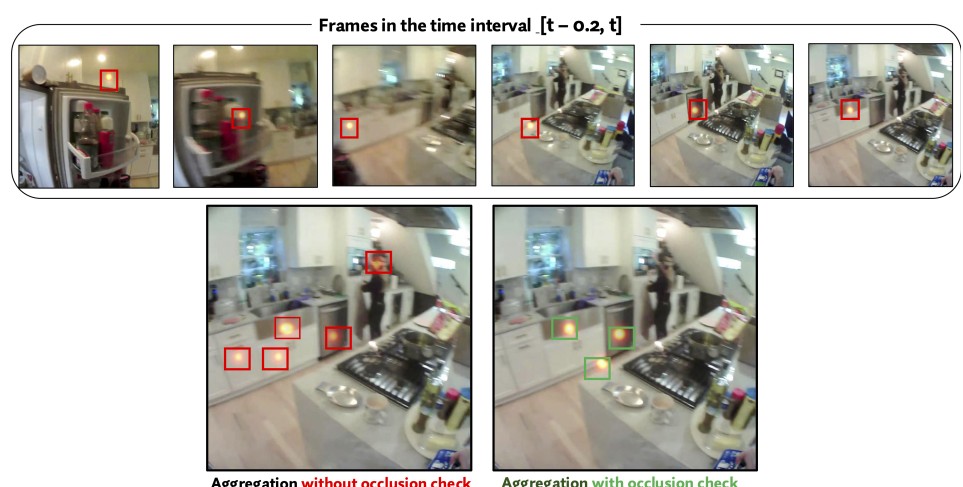

Figure 10: An example of gaze aggregation on the final image frame is shown. **Bottom left**: Result without occlusion check. **Bottom right**: Result with occlusion check applied. The fridge is absent in the final frame, indicating significant occlusion. Therefore, the gaze points from the first two frames, which are associated with the fridge, should not be overlaid on the final frame as the object is no longer visible.

## 8.3 OTHER EXPERIMENTS

In this section, we provide some details about the ablation studies conducted during the course of our study. We first explore the impact of the size of the gaze points used in the formation of the heatmap on the model performance. In addition, we employ a self-attention block to the base model without gaze to assess the result comparison with the gaze-augmented model. We also utilize gaze in text form to investigate whether our need to use gaze based images is necessary or not, followed by an investigation of the impact of changing observational and prediction horizons on model performance.

### 8.3.1 CHANGES IN THE SIZE OF THE OVERLAYS OF THE GAZE POINTS

In our exploration of gaze-augmented models, we investigated two primary approaches: the singular gaze model and the aggregated gaze model. The singular gaze model utilizes only one gaze point to

Table 5: Effect of the size of the gaze point overlays on model accuracy

| Gaze | Semantic | Meteor | Rouge-L (↑) | | |
|------|----------|--------|-----------|--------|---------|
| Model | Score(↑) | Score(↑) | Precision | Recall | F-score |
| Singular | 0.7316 | 0.4501 | 0.4822 | 0.5309 | 0.5054 |
| Singular (larger overlays) | 0.7683 | 0.4922 | 0.5123 | 0.5508 | 0.5304 |
| Aggregated | **0.7826** | 0.5033 | **0.5193** | **0.5644** | **0.5405** |
| Aggregated (larger overlays) | 0.7816 | **0.5060** | 0.5128 | 0.5556 | 0.5330 |

Table 6: Comparison of base model with attention block against gaze augmented models

| | Semantic | Meteor | Rouge-L (↑) | | |
|------|----------|--------|-----------|--------|---------|
| Model | Score(↑) | Score(↑) | Precision | Recall | F-score |
| Base | 0.6525 | 0.4075 | 0.4335 | 0.4301 | 0.4318 |
| Base (w self-attention) | 0.6701 | 0.4215 | 0.4292 | 0.4508 | 0.4393 |
| Aggregated gaze | **0.7826** | **0.5033** | **0.5193** | **0.5644** | **0.5405** |

create the binary heatmap and the gaze-overlaid image. To assess the impact of the gaze point size on model performance, we conducted experiments where we increased the size of the overlaid gaze points on the heatmap. Our findings indicated that performance improved with larger gaze point overlays in the singular gaze model.

In contrast, the aggregated gaze model, which combines multiple gaze points and includes an occlusion check, did not exhibit significant performance gains when the size of the overlaid points was increased. Table 5 highlights the results of this experiment. The results suggest that the aggregated gaze model's incorporation of multiple gaze points and an occlusion check is more effective than merely enlarging individual gaze points in the singular model.

### 8.3.2 INCLUSION OF SELF-ATTENTION BLOCK IN BASE MODEL WITHOUT GAZE

Building on the insights from the previous section, a natural question arises: what happens if we employ a self-attention block in the base model, providing the largest possible overlay—the entire image itself? In this experiment, we integrated a self-attention block into the base model without gaze. The features obtained from the ViT were passed into the self-attention block, where the queries, keys, and values were derived from these image features.

The objective of this modification was to investigate the effects of using a full image overlay as the query input to the attention block in the base model. The resultant features from the self-attention block were then forwarded to the perceiver resampler for further processing. Our results showed that while the performance of the base model improved with the inclusion of the self-attention block, it still remained below that of the gaze-augmented models, as shown in Table 6. This indicates that although leveraging the entire image enhances the base model's capabilities, it does not fully match the performance benefits achieved by incorporating eye gaze signal and using gaze-regularized attention mechanism.

### 8.3.3 USING GAZE IN TEXT FORM

In our studies, we converted gaze data from coordinate text form into heatmaps, which were then overlaid on scene images to create a more visual representation. This transformation allows us to highlight important visual regions and aligns more closely with how humans perceive spatial information. (Laeng et al., 2014). To conduct a sanity check and assess whether using gaze data in visual form is more suitable than using gaze in text form , we trained a gaze-augmented model that utilizes gaze coordinates as text input. Let $\phi_{gaze,text}$ represent the gaze augmented VLM which aims to model the likelihood of the fine grained text descriptions (represented by $\ell_i$) when eye gaze

Table 7: Effect of using gaze information in text form and comparison with other models

| Model | Semantic Score($\uparrow$) | Meteor Score($\uparrow$) | Rouge-L ($\uparrow$) | | |
| --- | --- | --- | --- | --- | --- |
| | | | Precision | Recall | F-score |
| Base | 0.6525 | 0.4075 | 0.4335 | 0.4301 | 0.4318 |
| Aggregated gaze(in text form) | 0.7021 | 0.4428 | 0.4642 | 0.4621 | 0.4630 |
| **Aggregated gaze** | **0.7826** | **0.5033** | **0.5193** | **0.5644** | **0.5405** |

information in the form of text coordinates $\{(x,y)\}$ is also provided:

$$\phi_{gaze,text}(\ell, \{I\}, \{(x,y)\}) = \prod_{i=\tau_o+1}^{\tau_o+\tau_a} p(\ell_i \mid \ell_{<i}, I_{\leq\tau_o}, \{(x,y)\}_{\leq\tau_o}) \qquad (14)$$

Our results indicated that using gaze data in text form improved performance compared to the base model without gaze. However, it still fell short of the performance achieved by the aggregated gaze-augmented model (as shown in Table 7). This experiment demonstrates that incorporating gaze information as a signal is essential. However, utilizing gaze in conjunction with the gaze-regularized attention mechanism significantly enhances model performance

### 8.3.4 CHANGING PREDICTION AND OBSERVATION HORIZONS

To investigate the effect of sequence length on model performance, we conduct experiments by adjusting the prediction horizon ($\tau_p$) and observation horizon ($\tau_o$). First, we extend the prediction horizon from 2 to 5 seconds to evaluate both the base and gaze-augmented models (using the aggregated gaze model with occlusion check). As shown in Table 8, the gaze-augmented model consistently outperforms the base model, even with a longer prediction horizon reinforcing the intuition that gaze is important to predict intentions and future actions. We also reduce the observation horizon to 3 seconds while keeping the prediction horizon fixed at 2 seconds. The gaze-augmented model again outperforms the base model (Table 9). Interestingly, the shorter observation horizon shows slightly increased performance compared to the main results reported with a longer observation horizon. This could indicate a potential improvement of our approach, by accounting for the visual working memory humans utilize.

Table 8: On increasing the prediction horizon (from 2 seconds) to predict future actions up to 5 seconds, the gaze-augmented model is able to outperform the base model.

| Model | Semantic Score($\uparrow$) | Meteor Score($\uparrow$) | Rouge-L ($\uparrow$) | | |
| --- | --- | --- | --- | --- | --- |
| | | | Precision | Recall | F-score |
| Base | 0.6297 | 0.3972 | 0.4393 | 0.4065 | 0.4220 |
| **Aggregated Gaze** | **0.7519** | **0.4891** | **0.5332** | **0.5410** | **0.5386** |

Table 9: On decreasing the observation horizon (from 5 to 3 seconds) to predict future actions up to 2 seconds, the gaze-augmented model is again able to outperform the base model.

| Model | Semantic Score($\uparrow$) | Meteor Score($\uparrow$) | Rouge-L ($\uparrow$) | | |
| --- | --- | --- | --- | --- | --- |
| | | | Precision | Recall | F-score |
| Base | 0.6716 | 0.4284 | 0.4544 | 0.4627 | 0.4581 |
| **Aggregated Gaze** | **0.7855** | **0.5132** | **0.5399** | **0.5523** | **0.5457** |

### 8.4 LIMITATIONS

In this section, we highlight the key limitations of our work, which is as follows:

- **Dataset Creation**: The dataset was created using an iterative prompt mechanism to refine a suitable template, which was then tested on a smaller subset of the data. Template approval was carried out by two individuals, but we acknowledge that the suitability of annotations can be subjective, potentially leading to varying results across different individuals. Due to the impracticality of manually verifying all annotations, we randomly checked 200 annotations from various videos in the dataset to ensure suitability. While mistakes were found and rectified, we acknowledge that the annotations may still contain imperfections. Additionally, comparisons with different modalities (e.g., audio) were not feasible, as not all gaze-equipped videos in the dataset had accompanying modalities. As a result, we did not propose a new benchmark but focused on demonstrating the importance of including eye gaze as an additional modality in a VLM to enhance its predictive capabilities.

- **Inference Speed Analysis**: A detailed study comparing the real-time inference speed of the baseline model and the gaze-augmented model could further strengthen our findings. However, this analysis was not conducted due to the lack of appropriate equipment.

- **SOTA Comparison**: Since our dataset and problem definition differ from prior studies, a direct comparison with state-of-the-art (SOTA) methods was not performed. Nonetheless, we emphasize that the primary contribution of this work is not the introduction of a new dataset or benchmark but rather the demonstration of how incorporating eye gaze into VLMs can improve their predictive tasks compared to relying solely on traditional inputs.

## 8.5 TRAINING AND EVALUATION DETAILS

Both the base model and the gaze-augmented model were trained using two NVIDIA A800 40GB GPU cards. The base model required approximately 36-38 hours for training, while the gaze-augmented model took around 50 hours. The training utilized a batch size of 32 and a learning rate of $7e-5$. The vision encoder is pre-trained by OpenAI. Additionally, the tokenizer for the text annotations used was the OPT (Open Pre-trained Transformer) language model developed by Meta. To accelerate the training process, we employed Fully Sharded Data Parallel (FSDP), a technique that efficiently distributes model parameters and gradients across multiple GPUs, reducing memory usage and improving training speed. Data loading was managed using the WebDataset loader, and the dataset was converted to .tar files to align with the format required for integration with both the WebDataset loader and FSDP.

In terms of model complexity, the base model has approximately 944 million parameters. The gaze-augmented model with 5 layers contains about 996 million parameters while the gaze-augmented model with 1 layer has around 955 million parameters. Lastly, the model with 2 layers, which is the best performing model, has approximately 966 million parameters.

For evaluation, both models were assessed using the semantic transformer (SBERT) developed by Reimers & Gurevych (2019), along with ROUGE-L and Meteor scores. The inference time on the test set for the base model was approximately one hour, while the gaze-augmented model had a slightly longer inference time but remained comparable to that of the base model.

