# OpenReview forum: "Gaze-Regularized Attention for Human Action Prediction"
_ICLR.cc/2025/Conference — Submitted to ICLR 2025_

### Official Review · Reviewer_82GE · 2024-10-24

**Soundness:** 2
**Presentation:** 2
**Contribution:** 1
**Rating:** 3
**Confidence:** 5

**Summary:**

This study introduces a novel approach to improving VLMs for human action prediction by integrating eye gaze data into egocentric video analysis. The method involves generating gaze heatmaps from eye gaze coordinates and dynamically focusing the model's attention on regions highlighted by gaze patterns. A gaze-regularization mechanism ensures the model maintains attention on gaze-allocated areas, enhancing prediction accuracy. The approach achieved a nearly 13% improvement in the semantic score compared to baseline models (i.e. no gaze), demonstrating the effectiveness of incorporating gaze data into VLMs for action anticipation.

**Strengths:**

- The ablation study is adequate, with a lot of experiments.

**Weaknesses:**

Title, intro & Abstract:
•	I believe the title (as well as the intro) of the paper is misleading. If the task at hand to improve VLM performance on action prediction/anticipation than VLM should exist in the title. Otherwise, it reads like an action recognition work.
•	Also, the term "prediction" seems to be replaced with "anticipation."
•	The task is not "action" based as the output is a description (like a caption). I do not see any parser that takes the final description and receives the actions, the output description has multiple actions while the original GT has a single action and noun.
•	The first sentence of the related work section is also inaccurate: “Action prediction and activity forecasting tasks both aim to anticipate future human actions.” The term "forecasting" implies that future actions are known at the time of prediction. The anticipation definition provided by Ego4D indicates that future frames are unavailable during prediction. In prediction, future frames are processed to predict actions, object interaction locations, etc. The second sentence improves upon the first but uses the vague phrase "longer time." I recommend adopting the Ego4D definition, as the method was evaluated based on this benchmark.

Related work:
•	If the task is to improve VLMs for action prediction, then why there is no section discussing VLMs in this line?
•	Several works are cited as Arxiv, despite being accepted in different venues. The references in the related work section should be updated accordingly. This is crucial for ensuring comparisons with state-of-the-art (SOTA) methods, which is completely missing in this study.
•	It appears the authors have overlooked the winners of the Ego4D anticipation task from the past few years.
•	The statement, “However, in our study, we use eye gaze data as a direct signal for human action anticipation, building a gaze-regularized attention mechanism,” suggests that the authors have not considered the foundational works by James Rehg and his group, who have demonstrated that gaze signals can significantly enhance action anticipation and prediction in egocentric videos. It would be beneficial to review the ETRA conference and GAZE workshops. Relevant examples include:
[A] Ozdel et al. Gaze-Guided Graph Neural Network for Action Anticipation Conditioned on Intention
[B] Fathi et al. Learning to Recognize Daily Actions Using Gaze
[C] Ozdel et al. Gaze-Guided Graph Neural Network for Action Anticipation Conditioned on Intention
[D] Zhang et al. Can Gaze Inform Egocentric Action Recognition?
•	On the other hand, I do not see any advantage of using the gaze signal which exists as a modality in the proposed method (also in test time) while several methods can predict the gaze and actions simultaneously and use the gaze to improve the action prediction. It would be great if the authors clarify this.

Annotation and Dataset Concerns:
•	It is crucial to note that since the proposed method relies on gaze signals as a modality—differing from other studies performs action prediction/anticipation—the dataset split utilized is significantly smaller. Limiting experiments to one dataset, which is substantially smaller than the most recent action prediction SOTA, poses a disadvantage. A more realistic approach would involve using a gaze target detector and incorporating its output, as in real-life applications, gaze location will not be provided as ground truth.
•	How did you evaluate the text descriptions manually for such a large number of images? What was the duration of the evaluation process? How many individuals assessed each image, and what was the consensus among them? Are there any statistics available regarding this consensus?

Method:
•	The "GAZE-REGULARIZED ATTENTION BLOCK" bears a strong resemblance to the work of [E], which uses object embeddings instead of gaze, nevertheless, its technical novelty is unclear.
[E] Thakur et al., Leveraging Next-Active Objects for Context-Aware Anticipation in Egocentric Videos, WACV 2024.
•	The evaluation metrics employed diverge significantly from those suggested by Ego4D. The task encompasses more than mere action recognition and should not be cast to captioning or if it is captioning then why mentioning the action recognition in the title and related work and thought the paper?

Evaluation:
•	Ablation study is adequate.
•	Comparisons with SOTA are missing. What about using different large models? Why only flamingo?
•	The task defined by Ego4D where the noun, action, and time of contact are to be predicted, was significantly changed. Consequently, comparisons with other methods are harder. However, the current results are not justifying the performance of the proposed method is significant or not as it is just ablated but not compared with other approaches. The results demonstrate that gaze matters, however, this is something we already know from the literature.

Overall & justification of the score:
HCI and computer vision studies have repeatedly shown that gaze is an important indicator of both current and future actions. This study presents similar findings. However, the method of incorporating the gaze signal is not novel even if it might be the first time for a VLM case while it was not tested on several ones, but flamingo was used.
The limitations of the approach were not discussed. Additionally, both the ground truth and predictions were generated by VLM, which raises concerns. The findings are unsurprising, and the technical novelty is unclear. There are no comparisons with other papers using Ego4D, which may be due to significant changes in the ground truth. However, it seems possible to inject VLM into other methods and perform a comparison. This possibility was never been discussed.

**Questions:**

- Why did you change the definition of the existing action prediction (and/or anticipation) task, which is well described in Ego4D? The GT has been significantly changed and the produced output is also very different, what are the advantages of them wrt to Ego4D's SOTAs' implementation? And do you think what you perform in the paper is action recognition?
- What is the advantage of using the gaze signal directly? Would not it be better to regress the gaze simultaneously?
- What are the limitations of your work?

---

> ### Author Response · Authors · 2024-11-22
>
> We sincerely thank the reviewer for their valuable feedback and for providing additional references, which will be instrumental in guiding our future research and work.
> We agree with the reviewer’s suggestion and have decided to include "VLM" in the title to give readers a clearer understanding of the paper’s focus at a glance. The new title is :**VLMs with Gaze-Regularized attention for
> human action prediction**.
> Additionally, we acknowledge that the terms "prediction" and "anticipation" have been used interchangeably in the paper. Based on the definitions provided by Ego4D and standard usage, we have ensured consistent terminology in the final revision, incorporating all suggested changes.
>
> ### Redefining the problem statement
> We have re-defined the problem setting to focus on fine-grained actions, as they are inherently more informative than coarse-grained actions. This aligns with the structure of most instruction sets or manuals, which typically provide detailed, step-by-step guidance rather than high-level action or verb tuples. The primary aim of our study was to demonstrate, through empirical results, that for tasks requiring predictive capabilities, relying solely on vision in VLMs is insufficient. By introducing an additional modality—eye gaze—we achieved more accurate predictions of future activities. Specifically, fine-grained annotations were significantly enhanced, providing better descriptions of actions, objects, and potential trajectories when gaze data was incorporated. Additionally, we updated the definition of suitable instruction prompts based on the belief that they should encompass key components: the objects involved, the actions performed by the camera wearer or individuals in the scene (in the context of egocentric vision), and the trajectories taken. These factors were carefully considered while designing prompts for the VLMs to generate accurate and comprehensive annotations.
>
> ### Gaze as a direct signal
> Gaze is not utilized as a strict rule but rather as a preference, allowing for a balance between the natural attention mechanism provided by eye gaze and the model-driven attention through the attention block. Furthermore, with proper system calibration, it is possible to obtain the original gaze ground truth directly, eliminating the need for reliance on a predicted gaze ground truth.
>
> ### Limitations of our work
> We would like to thank the reviewer for bringing to our attention that the limitations of our study were not mentioned in the original submission. Some of the limitations of our work are stated below:
> * **Dataset Creation**: The dataset was created using an iterative prompt mechanism to refine a suitable template, which was then tested on a smaller subset of the data. Template approval was carried out by two individuals, but we acknowledge that the suitability of annotations can be subjective, potentially leading to varying results across different individuals. Due to the impracticality of manually verifying all annotations, we randomly checked 200 annotations from various videos in the dataset to ensure suitability. While mistakes were found and rectified, we acknowledge that the annotations may still contain imperfections. Additionally, comparisons with different modalities (e.g., audio) were not feasible, as not all gaze-equipped videos in the dataset had accompanying modalities. As a result, we did not propose a new benchmark but focused on demonstrating the importance of including eye gaze as an additional modality in a VLM to enhance its predictive capabilities.
> * **Inference Speed Analysis**: A detailed study comparing the real-time inference speed of the baseline model and the gaze-augmented model could further strengthen our findings. However, this analysis was not conducted due to the lack of appropriate equipment.
> * **SOTA Comparison**: Since our dataset and problem definition differ from prior studies, a direct comparison with state-of-the-art (SOTA) methods was not performed. Nonetheless, we emphasize that the primary contribution of this work is not the introduction of a new dataset or benchmark but rather the demonstration of how incorporating eye gaze into VLMs can improve their predictive tasks compared to relying solely on traditional inputs.

---

### Official Review · Reviewer_37M3 · 2024-11-03

**Soundness:** 3
**Presentation:** 2
**Contribution:** 2
**Rating:** 5
**Confidence:** 3

**Summary:**

The paper presents an innovative approach that utilizes gaze as an attention map to guide visual features within a Vision Language Model (VLM) for human action prediction in an egocentric context. In this method, gaze data collected across a sequence of frames is aggregated into a gaze heatmap. This heatmap is divided into patches, with a probability distribution defined over each patch to reflect the gaze intensity. The heatmap is overlaid on the original RGB images, which are then processed by a Vision Transformer (ViT) to generate gaze-aligned features.

In the attention mechanism, these gaze features serve as queries, while the RGB image features extracted by the ViT act as keys and values. The model predicts attention weights by aligning the queries with these keys and values, facilitating attention guided by gaze patterns. To encourage alignment, a Kullback-Leibler Divergence (KLD) loss is applied, ensuring that the VLM’s attention distribution closely matches the gaze-derived distribution. This setup allows the model to dynamically focus on regions of visual significance while incorporating human gaze cues, which enhances the interpretability and relevance of its predictions for egocentric action anticipation.

**Strengths:**

The authors present a straightforward yet effective way to integrate gaze data into a Vision Language Model (VLM), guiding the model's attention to human gaze-directed areas. By aligning machine attention with human perceptual cues, the approach enhances the model’s understanding of intent and supports more accurate action prediction, especially in dynamic, egocentric scenarios.
Their results indicate a substantial improvement over the baseline model lacking gaze-based guidance, with a notable 13% increase in accuracy. The paper also includes a thorough experimental comparison with the baseline model and an ablation study exploring various methods of incorporating gaze as an attention mechanism.
One particularly impressive aspect of the approach is the use of gaze features, rather than relying solely on heatmaps. By optimizing the attention distribution to reflect gaze patterns, the model effectively benefits from human guidance in identifying critical visual areas. This method, while relatively simple, proves both intuitive and powerful in enhancing action anticipation accuracy.

**Weaknesses:**

While the paper is well-written and demonstrates clear improvements over a basic VLM baseline, I find the experimental section overly focused on their proposed approach without sufficient comparisons to existing non-gaze-based methods on the Ego4D dataset. For a more comprehensive evaluation, it would have been beneficial to see comparisons with alternative models that perform action prediction on Ego4D using methods other than gaze. Examples such as PALM: Predicting Actions through Language Models and MMG-Ego4D: Multi-Modal Generalization in Egocentric Action Recognition could provide useful context, though they are just suggestions. Such comparisons would show how the proposed approach stands relative to state-of-the-art action prediction methods in similar settings rather than only against a weaker baseline.
Additionally, I see potential in the gaze embedding component as a general enhancement for VLMs, which could be more compelling if evaluated across a range of tasks or compared with other gaze-based approaches. For instance, examining how the model compares to Voila-A: Aligning Vision-Language Models with User's Gaze Attention could offer insights into its performance in similar gaze-enhanced VLM setups.
Considering that gaze data collection requires specialized equipment, it would strengthen the argument if the authors could demonstrate that gaze embeddings lead to clear accuracy gains over non-gaze-based methods. Testing with other vision-language models, such as LXMERT, could showcase the potential generalizability and robustness of the gaze embedding across models.

Minor Suggestions: Adding a grid overlay to Figure 3 would help clarify the construction of patches for the reader, improving the paper’s visual clarity in illustrating the method.

**Questions:**

1- Equation 4 Clarification: The author points out that the denominator refers to all pixels in the image rather than just those in the patch. This choice may impact how attention weights are normalized across different regions and could benefit from further explanation. Why does the model use global pixel summation instead of patch-specific summation? Additional clarification on this point would be helpful.

2- Gaze-Overlaid Image Processing: The statement, "We adopt a methodology similar to that used for RGB images to process gaze-overlaid images with a Vision Transformer (ViT) for feature extraction," is confusing. The authors did not explain what exactly was used for RGB images. A more detailed description of what is meant by "similar to that used for RGB images" would enhance clarity.

3- Study of Temporal Variations: Given the paper's proposal to use gaze-guided features for action prediction, a deeper analysis of how the model performs over varying input and prediction times would be valuable. I think including results form Table 8 and 9 from supp material into the main paper could strengthen the paper further.

4- Model Speed and Real-Time Performance: Since action prediction in an egocentric setting often requires near real-time responses, it would be useful to provide details on the model's runtime. An analysis of inference speed, particularly for real-time applications, would strengthen the model's practical applicability.

5- Construction of Gaze Feature Images: More information on how gaze feature images are constructed is essential for reproducibility. Exploring alternative ways of generating these features could offer insights into the method's sensitivity to gaze representation. Additionally, a study on the distribution of the Gaussian overlay on gaze data could provide insights into how different spatial representations impact model performance. Testing variations, such as adjusting the Gaussian’s spread or opacity, could highlight the robustness of the gaze feature representation

---

> ### Author Response · Authors · 2024-11-22
>
> Thank you for the suggestions, as well as taking the time to provide valuable feedback to enhance our research.
> We will include PALM as a reference in our paper, as it also served as an inspiration for our work with egocentric vision. However, a direct comparison with PALM would be inappropriate, as their focus is on coarse-grained activity sequences, which they handle effectively, whereas our work emphasizes fine-grained activity recognition.
> Regarding MMG-Ego4D, we agree that a comparison across different modalities would provide a stronger indication of the effectiveness of gaze as a modality. However, for our original dataset, not all videos contain consistent modality information—some include only gaze, others have both gaze and audio, and some are limited to gaze alone. This inconsistency is a limitation of our work, and we will include it in the paper following further feedback from the reviewers.
> In addition, the grid overlay would indeed enhance the visibility of the method but since readers could also get confused as what the actual gaze-overlaid image and binary heatmap image looks like (which does not include the grid), we have decided to leave it untouched at the moment.
> ### Significance of global summation
> To calculate the gaze distribution, we use a binary heatmap image where gaze-occupied regions are illuminated (white pixels), while the rest of the image is blacked out (black pixels). The variable
> p can only take values of 0 (for blacked-out pixels) or 1 (for illuminated pixels).
> We opted for a global summation approach, as it reflects the proportion of total gaze attention (white pixels) present in each designated patch. This ensures that attention is distributed based on these proportions. If a patch-level summation were used as the denominator, every patch with gaze would be assigned a value of 1, and our approach will not be usable for attention regularization.
> For example, using a global summation as the denominator: if patch 1 has 5 illuminated pixels, patch 2 has 5 illuminated pixels, and the rest of the image is blacked out, the calculated proportion for patch 1 and patch 2 would be 0.5 each, while all other patches would have a value of 0. This effectively indicates how attention should be divided across the image patches, maintaining a proportional representation of gaze distribution.
> ### Study of temporal variations
> We agree that the study of temporal variations is an important part of the paper and is interesting, however, we decided not to include the results in the main paper due to space constraints, as well as the fact that we were only able to provide an educated intuitive explanation to the empirical results for the variations in observational/anticipation time.
> ### Change in overlay size and opacity
> We have provided results for the case where the size of the Gaussian overlay was increased (refer to Tables 5 and 6). For the base model, increasing the overlay size resulted in a noticeable performance improvement, whereas the improvement for the aggregated gaze model was minimal. Regarding opacity, we conducted an experiment with fully transparent overlays and included the regularizer as well. The semantic score for this experiment was **0.671985**, representing a slight improvement over the base model but still below the gaze-augmented models with gaze overlays visible.
> **Note**: This model was trained under identical conditions using gaze-overlaid images with transparent overlays. This setup differs from the experiment mentioned in Comment 2, where a gaze-augmented model trained with gaze-overlaid images was tested with only RGB images (without overlays) to assess its performance.

---

> > ### Comment · Reviewer_37M3 · 2024-11-27
> >
> > Thank you for your response and the provided verifications. While I appreciate the idea of introducing an easy-to-integrate module for VLMs, I agree with the other reviewers that conducting a comprehensive comparison with existing methods would significantly strengthen the paper and support its claims more effectively.
> >
> > Regarding the comparisons, I find your argument unconvincing that fine-grained predictions make it unsuitable to compare your work with related methods that focus on coarse predictions. I would expect that fine-grained predictions still encompass data relevant to coarse predictions, making such comparisons feasible and insightful.
> >
> > Additionally, I would like to learn more about the runtime performance of your model, as this is a critical factor in evaluating its practical utility.
> >
> > I am also a bit puzzled about the answer provided to reviewers eGw6 and dRXv
> >
> > Model	Semantic Score	Precision	Recall	F-Measure
> > Base w RGB image	0.6525	0.4335	0.4301	0.4318
> > Base w Gaze-overlaid image	0.6873	0.4332	0.4553	0.4435
> > Base w RGB image	0.7826	0.5193	0.5644	0.5405. >>> I think here there is also a typo in the name of the exp?
> >
> > Model	Semantic Score	Precision	Recall	F-Measure
> > Base Model	0.6525	0.4335	0.4301	0.4318
> > Aggregated gaze w RGB queries	0.7368	0.4958	0.5113	0.5034
> > Aggregated gaze w gaze overlaid queries	0.7826	0.5193	0.5644	0.5405

---

> ### Author Response · Authors · 2024-12-01
>
> We would like to thank the reviewer for pointing out the typing error, which has been corrected in the previous comments.
>
> Regarding the runtime, we would like to clarify that the evaluation was conducted using RGB images for the base model and RGB+gaze-overlaid images for the aggregated gaze model. On average, both models took approximately 1.7 to 2 seconds to process a sequence. This runtime was calculated based on the total time required to run the test set divided by the total number of samples in the test set.
>
> For the aggregated gaze model, the incorporation of an occlusion check using forward and backward optical flow adds an additional 2-3 seconds per sequence, resulting in a slightly slower runtime compared to the base model. However, this occlusion check improves the quality of predictions, as reflected in the semantic scores.
>
> If the occlusion check (optical flow with consistency) is excluded during test time, the semantic score decreases from **0.7826 to 0.7702**, highlighting its contribution to model performance. Additionally, if the occlusion check is removed during training, the semantic score drops further to **0.7616**. These results emphasize the importance of consistency checks in improving the model's semantic understanding. In addition, we aim to present the gaze-regularized framework as an adaptable component that can be integrated into Vision Language Models (VLMs). This framework is designed to work seamlessly with transformer-based architectures that utilize separate vision and language modules, which we think will make it suitable to other VLMs.

---

### Official Review · Reviewer_dRXv · 2024-11-04

**Soundness:** 3
**Presentation:** 3
**Contribution:** 2
**Rating:** 3
**Confidence:** 4

**Summary:**

The paper presents a gaze-guided vision-language model (VLM) for fine-grained human action prediction. The authors leverage the gaze data present in the data set Ego4D as input signal to create gaze-augmented images and feed them into a vision transformer (ViT). The resulting tokens are used as queries in an attention block, where similarly processed tokens of RGB images are used as keys and values. The attention block is regularized to align the transformer’s attention with the human gaze heatmaps through Kullback-Leibler divergence. The authors compare their model to a baseline model without gaze guidance and achieve a ~13% improvement in performance.

**Strengths:**

The paper is well written and presents good experiments and ablation studies to validate their method. The method is described clearly and reproducibility will be ensured by providing the relevant code. In general, action prediction is a valuable task for human-machine interaction, which the authors present as their motivation.

**Weaknesses:**

- The significance of the method for the ICLR community remains unclear, as the topic mainly belongs to the field of human-computer interaction (HCI) and does not incorporate any representation learning, which is clear as no ICLR papers are used as reference. Hence, it might make sense to redirect this paper to a more HCI specific conference.

- Guiding a model’s attention with eye gaze is not novel (Min & Corso, 2020; Awale & Sarikaya, 2022) +  [N1, N2, N3] and usually includes a gaze prediction model instead of requiring gaze at inference time. This key limitation of the method has been described as a feature, because “eye-gaze can be easily obtained and should be used as a direct signal” (L1.41), which remains questionable.

- The authors only compare their method to a simpler baseline model instead of other methods described in the literature, e.g. (Min & Corso, 2020).

- Several key references are missing, e.g. vision transformer [N4], similarly for ChatGPT, GPT-V4, and ShareCaptioner. In general, I would suggest adopting a more comprehensive citation style and finding references for claims, such as “VLMs have the potential to facilitate human-machine interaction” (L.40) or “[these scores] are widely employed to assess text similarity” (L.410).

- Abbreviations are not used consistently, i.e. vision transformer (ViT) is almost always written in full, same as vision-language model (VLM).

[N1] Ruohan Zhang, Zhuode Liu, Luxin Zhang, Jake A Whritner, Karl S Muller, Mary M Hayhoe, and Dana H Ballard. (2018). Agil: Learning attention from human for visuomotor tasks. In Proceedings of the european conference on computer vision (eccv). 663–679.

[N2] Hu, Z., Schmitt, S., Haeufle, D., & Bulling, A. (2024). GazeMotion: Gaze-guided Human Motion Forecasting. arXiv preprint arXiv:2403.09885.

[N3] Özdel, S., Rong, Y., Albaba, B. M., Kuo, Y. L., Wang, X., & Kasneci, E. (2024). Gaze-Guided Graph Neural Network for Action Anticipation Conditioned on Intention. In Proceedings of the 2024 Symposium on Eye Tracking Research and Applications (pp. 1-9).

[N4] Dosovitskiy, A., Beyer, L., Kolesnikov, A., Weissenborn, D., Zhai, X., Unterthiner, T., Dehghani, M., Minderer, M., Heigold, G., Gelly, S., Uszkoreit, J., & Houlsby, N. (2020). An Image is Worth 16x16 Words: Transformers for Image Recognition at Scale. International Conference on Learning Representations.

**Questions:**

Based on the ablation studies it remains unclear why the use of gaze overlaid images (G) as queries in the attention block is important. The attention-guidance through KL divergence seems to be the main reason for the performance increase, as performance drops even below baseline performance if lambda is set to 0 (Table 2). Therefore, I would suggest performing a more detailed analysis and potentially removing this additional input step, which in turn would make it easier to remove the need for eye gaze signals at inference time.

Since delta is set to 200 ms and the eye gaze is sampled at 30 fps, the maximum number of gaze points considered in one prediction is 6 (and this is most likely during a saccade). Further, the images considered as input are downsampled to 1 fps. I believe it would be interesting to evaluate whether increasing the amount of gaze points (delta) up to the 30 points corresponding to one image frame, can increase the performance of the model. Additionally, it would be interesting to see how the data is distributed, i.e. how many gaze points are within the selected distance discrepancy threshold (eta) and why delta has been selected to be 200ms.

---

> ### Author Response · Authors · 2024-11-22
>
> Thank you for your suggestions and for taking the time to provide us with the feedback. We also appreciate the references that were provided to aid with our study.
> We have ensured consistency with the abbrievations in the final paper, and also included references for ShareCaptioner, GPT-4V and also other places where we think we have made claims without providing sufficient justification.
> ### Significance of method
> While we acknowledge that the paper could be categorized as primarily HCI-focused, we believe our proposed research also aligns with the broader scope of hybrid AI systems. Our work draws inspiration from the human visual system and incorporates these principles into a VLM. Furthermore, we see significant potential for this gaze-augmented VLM in applications such as assisted living environments, with implications for robotics, autonomy, and planning. Notably, both of these areas are listed among the potential topics for the ICLR 2025 call for papers.
> ### Eye gaze as signal
> We greatly acknowledge the foundational work by researchers in gaze prediction and estimation, as well as efforts that integrate gaze prediction with human action recognition through multitask objectives. However, our approach uses gaze as a direct signal, not as a strict rule, but as a preferred method to balance model-derived attention with gaze as a natural attention mechanism. Importantly, when the system is correctly calibrated, gaze ground truth can be directly obtained rather than relying on predicted values.With the growing adoption of VR/AR headsets and commercially available eye-tracking glasses, obtaining precise gaze data is becoming increasingly accessible. This trend highlights the practicality of our approach, as such hardware advances make gaze data easier to obtain and integrate into AI systems.
> ### Comparison with literature
> The literature discussed primarily focuses on gaze prediction and activity recognition. However, a direct comparison with our work would be inappropriate, as our approach diverges significantly in scope and objectives. Specifically, we aim to generate fine-grained and descriptive text annotations that detail actions, objects, and their trajectories while leveraging eye-gaze as a direct signal. This contrasts with existing work, which often relies on coarse-grained annotations for activity recognition and uses gaze primarily as supervision for gaze prediction. Our goal is to provide further empirical evidence on the under-explored potential of utilizing gaze as a direct signal or an additional modality within the context of a VLM. By doing so, we seek to predict future actions in the form of detailed, fine-grained annotations, moving beyond the traditional reliance on RGB images or video clips alone.
> ### Ablation studies with more gaze points
> Due to constraints in time and resources, we were unable to conduct experiments using 30 gaze points. However, to explore the impact of using a larger number of points (or a longer aggregation duration), we trained an aggregated gaze model utilizing 12 frames instead of 6. The results of this experiment are provided below
> | Aggregated Points            | Semantic Score | Precision | Recall | F-Measure |
> | ---------------- | -------------- | --------- | ------ | --------- |
> | 6 | 0.7826         | 0.5193    | 0.5644 | 0.5405    |
> | 12 | 0.7808         | 0.5143    | 0.5541 | 0.5334 |
>
> The negligible difference between the two scenarios can likely be attributed to occlusion checks, which may limit the total number of usable gaze points. In cases with minimal occlusion, the slight decrease in performance can be compared to that observed in an aggregated gaze model with larger overlays, where performance also decreased slightly. This result suggests that utilizing an excessive number of gaze points could potentially confuse the model. However, as this explanation is currently based on intuition rather than extensive analysis, it was not included in the main paper.
> ### Significance of the use of gaze overlaid images
> An experiment was conducted to evaluate the performance of the gaze-augmented model during testing when only RGB images (without gaze-overlaid images) were provided as input. This was done to assess how the previously trained gaze-augmented model performs in the absence of explicit gaze overlays. The results are provided below:
> | Model            | Semantic Score | Precision | Recall | F-Measure |
> | ---------------- | -------------- | --------- | ------ | --------- |
> | Base Model | 0.6525         | 0.4335    | 0.4301 | 0.4318    |
> | Aggregated gaze w RGB queries | 0.7368         | 0.4958    | 0.5113 | 0.5034    |
> | Aggregated gaze w gaze overlaid queries| 0.7826         | 0.5193    | 0.5644 | 0.5405    |
>
> This shows that during inference, if gaze-overlaid images are not used, then there is a drop in performance (approximately 5 percent).

---

> > ### Comment · Reviewer_dRXv · 2024-11-25
> >
> > Thank you for the additional experiments. However, both do not alleviate my concerns. It is still unclear what the gaze distribution looks like, i.e. whether the 6 or 12 gaze points are mainly fixations or parts of multiple or one saccade.
> >
> > More importantly, for the significance of gaze overlaid images, I was interested in a model that is trained only with gaze guidance through KL divergence, instead of the additional gaze overlaid images (G). Here the opposite has been implicitly presented when lambda is set to 0 (Table 2), where we see that a model that only uses G performs worse than the baseline. So my intuition would be that a model with regularization but without G still achieves the best performance, indicating that G is not important. Without G the need for gaze as input signal is removed, however, also the contribution of the method is even smaller, as gaze-guidance has already been done before in various other tasks. Overall, my concerns remain and I will not change my score.

---

> ### Author Response · Authors · 2024-11-25
> **Further clarification**
>
> Dear Reviewer dRXv,
>
> Thanks for your response.
>
> It seems that some effort is still needed to illustrate the importance of gaze-overlaid images.
>
> To further clarify, we elaborate on the two sides of gaze-overlaid images.
>
> First, gaze-overlaid images are important to facilitate the learning. Say, if we do not include gaze-overlaid images in the input, then the encoder (before the attention layer) that generates queries has to memorize what was happening in the past, so to produce queries that generate the desired attention pattern. However, since one history (without knowing the gazes) can correspond to multiple futures, then the encode has to learn an average, which decreases the learning efficiency. So it is important to include gaze-overlaid images, in such cases, the encoder does not need to memorize or guess what has happened, since what would happen in the future can be inferred to some extent already from the gaze pattern in the overlaid images.
>
> On the other side, gaze-overlaid images provide more information than the original images, thus increasing the chances of overfitting. This is observed in our experiment where the regularization is disabled. Thus, by adding gaze regularization, we can efficiently prevent overfitting to certain gaze-overlaid patterns and increase the generalization.
>
> By combining both gaze-overlaid images and gaze-regularization in the attention, our method can facilitate learning efficiency, by providing gaze information of the past to promote intention estimation, and improve the effectiveness and generalization, by applying gaze regularization to focus on the relevant scene information instead of short-cuts in the gaze patterns.
>
> Furthermore, we would like to point out that, gaze information is used in other works, but to our best knowledge, this is the first time to use gaze to regularize the attention maps in transformers (LLMs).
>
> We hope this clarification can help elaborate on the reasonings for our design choice and the underlying contributions we have made to the topic of using LLMs for ego-centric fine-grained human action prediction.
>
> We appreciate your time to reconsider our work and please let us know if more information is needed to help with your final evaluation of our work.
>
> Thanks,
>
> The Authors

---

> > ### Author Response · Authors · 2024-12-01
> >
> > ### Regularized model without gaze-overlaid images
> > We thank the reviewer for their feedback and clarifying the experiments required. In response to the reviewer's comments, we conducted an additional experiment to assess the impact of gaze-overlaid images on the model's performance. Specifically, during the training phase of this experiment:
> >
> >
> > * Gaze-overlaid images were not provided to the model.
> > * RGB queries were used instead of gaze-based queries.
> > * The gaze regularizer remained included as part of the training setup.
> >
> > The results of this experiment are as follows:
> >
> > | Model            | Semantic Score | Precision | Recall | F-Measure |
> > | ---------------- | -------------- | --------- | ------ | --------- |
> > | Base Model | 0.6525         | 0.4335    | 0.4301 | 0.4318    |
> > | Regularized model w/o gaze-overlaid images | 0.7505         | 0.5024    | 0.5331 | 0.5173    |
> > | Aggregated gaze w gaze overlaid queries| 0.7826         | 0.5193    | 0.5644 | 0.5405    |
> >
> > The aggregated gaze model with the gaze-based queries showed the best performance as compared to the model trained without gaze-overlaid images but with a regularizer.
> >
> > ###  Gaze Point Distribution and Fixation vs. Saccades:
> >
> > To address the reviewer’s concern regarding the gaze distribution, the attached table illustrates the distance distribution of gaze points relative to the final frame’s gaze point (frame 6). The distance metric reflects how much each corresponding gaze point (from frames 1–5) moves in pixels compared to the gaze point in frame 6. We acknowledge that the limited movement of gaze points within this dataset could suggest characteristics of fixations since fixations are generally associated with minimal movement as compared to saccades. However, the chosen threshold of 20 pixels was not directly intended to classify fixations or saccades. Instead, it serves as a practical measure to check for major occlusions in the gaze data and ensure that the model focuses on valid gaze regions. We experimented with multiple temporal thresholds for delta, including single gaze point , 6 gaze points and 12 gaze points. Among these, the model achieved the best performance with the 200 ms threshold (6 gaze points).
> >
> >
> > | Bin Range (Pixels) | Histogram Frequency |
> > |---------------------|---------------------|
> > | 0 - 10              | 346851              |
> > | 10 - 20             | 100730              |
> > | 20 - 30             | 49444               |
> > | 30 - 40             | 28819               |
> > | 40 - 50             | 18560               |
> > | 50 - 60             | 12693               |
> > | 60 - 70             | 8975                |
> > | 70 - 80             | 6650                |
> > | 80 - 90             | 4960                |
> > | 90 - 100            | 2101                |

---

### Official Review · Reviewer_eGw6 · 2024-11-04

**Soundness:** 2
**Presentation:** 4
**Contribution:** 2
**Rating:** 5
**Confidence:** 3

**Summary:**

This paper proposes a new framework that incorporates human eye gaze into the VLM training process for the task of fine-grained human action prediction. Compared to the VLM baseline (Flamingo) that is trained without any gaze information, the proposed approach achieves better performance. The dataset used for experiments is constructed using a subset of the Ego4D dataset.

**Strengths:**

The problem that is addressed in the paper (fine-grained action prediction in egocentric video) is well motivated and is an important problem that has great potential for various practical use cases.

The paper provides detailed descriptions of the dataset creation process and other details around the experimentation. This is is helpful for better understanding and reproducibility.

**Weaknesses:**

* Other baselines for fair comparison
There is only one baseline that is considered in the paper, which is a plain VLM model fine-tuned on the language and image without gaze. Since gaze itself is additional information that contains rich information about person's intention on its own, simply providing this info into the existing VLM pipeline would also increase the performance, even without the proposed gaze-regularized attention architecture.
In order to better evaluate the true value of the proposed method and to separate out the improvements coming from having access to additional information, additional baselines would be valuable. For example, if we simply provide gaze-overlaid images to base VLM, fine-tune and evaluate on them, how much would that help?

* Sensitivity to missing data
In real use case scenarios, gaze data can be frequently missing from the gaze tracker or have no gaze information at all. What is the impact of such cases on the model performance? Can an off-the-shelf saliency model be used in the absence of actual gaze? or would it be causing more harm than good if gaze data is missing? How big would be the gap if so?

* Generalizability
The whole idea in the paper is only validated using a single dataset, which is Ego4D, and even then it is a subset of it and small scale. Is the method generalizable to other scenarios? or is it optimized for only this use case? It needs to be evaluated on other scenarios/datasets.

**Questions:**

See above.

---

> ### Author Response · Authors · 2024-11-22
>
> We would like to thank the reviewer for taking the time to provide suggestions and valuable feedback for our research.
>
> ### Performance when simply gaze overlaid images are provided to base VLM
> To ensure that the performance improvement of our model is not merely due to the additional information provided by the gaze data, we conducted a sanity check. Specifically, we provided gaze-overlaid images as input to the base model instead of standard RGB images. The model was trained under the same conditions as the base model and gaze-augmented models. The results are provided below:
>
> | Model            | Semantic Score | Precision | Recall | F-Measure |
> | ---------------- | -------------- | --------- | ------ | --------- |
> | Base w RGB image | 0.6525         | 0.4335    | 0.4301 | 0.4318    |
> | Base w Gaze-overlaid image | 0.6873         | 0.4332    | 0.4553 | 0.4435    |
> | Aggregated Gaze w RGB image | 0.7826         | 0.5193    | 0.5644 | 0.5405    |
>
>  It is observed that using gaze overlaid images instead of RGB images improves the performance of the base model slightly, but it’s still significantly lower than the performance of the gaze augmented models.
>
> ### Sensitivity to missing data
> During the testing phase, we randomly selected images and corrupted their corresponding gaze-overlaid versions by removing the gaze points entirely. The corruption probability for each gaze-overlaid image was varied to evaluate the model's performance under different levels of corruption. If the corruption probability for an image in the observational sequence is 1, the image contains no gaze overlays (i.e., it is a standard RGB image). The results of this experiment are provided below.
>
> | Model            | Semantic Score | Precision | Recall | F-Measure |
> | ---------------- | -------------- | --------- | ------ | --------- |
> | 0 |  0.7826         | 0.5193    | 0.5644 | 0.5405    |
> | 0.2 | 0.7801         | 0.5191    | 0.5607 | 0.5390    |
> | 0.6| 0.7757         | 0.5160    | 0.5372 | 0.5264    |
> | 1| 0.7368         | 0.4958    | 0.5113 | 0.5034    |
>
> ### Off-the-shelf Gaze detector
> While it is possible to use an off-the-shelf gaze detector or predictor model, doing so would undermine the objective of leveraging gaze as a direct signal. To explore this further, we conducted an experiment where an additional self-attention mechanism was integrated into the original model (refer to Table 6 and Section 8.3.2). This experiment aimed to assess whether incorporating a self-attention mechanism designed to capture salient features would lead to a significant performance improvement. While the base model with the self-attention network showed some enhancement, the accuracy gap compared to the gaze-augmented models remained substantial.

---

### Meta-Review · Area_Chair_DjDu · 2024-12-19

**Metareview:**

This paper presents a method to integrate VLMs for human action prediction in egocentric scenarios with gaze information. Considering a subset of Ego4D data, gaze is used to selectively generate augmented images, which are then used to train a vision transformer. In practice, gaze data are adopted to drive the attention mechanism of the ViT model, showing to reach outstanding performance wrt state-of-the-art baselines.

As points of strength, this work is appreciated for the well motivated and important task it addresses, the quality and clarity of the writing, and the proper designed experiments.

Weak points are mainly related to the experimentation phase and ablations: it has been noted a lack of sufficient baselines for comparisons (only vanilla ViT), weak assessment of performance in actual use-case scenarios, where gaze data can be missing or noisy, scarce possibility to evaluate the generalization ability of the method since only a dataset is considered. Other issues discuss the novelty of the work, and several punctual requests for better/improved explanations or motivations of some methodological steps are also reported.

Authors tried to answer to all such remarks, but the reviewers resulted not fully convinced of the replies, and did not improve their judgments. In the end, the final scores were all below threshold (5, 3, 5, 3).

In these conditions, this paper cannot be considered acceptable for publication to ICLR 2025.

**Additional Comments On Reviewer Discussion:**

See above

---

### Decision · Program_Chairs · 2025-01-22

Reject